

# Global Fire Emissions Database burned-area dataset into Community Land Model version 5.0 – Biogeochemistry: Impacts on carbon and water fluxes at high latitudes

Hocheol Seo [1], Yeonjoo Kim [1]

[1]Department of Civil and Environmental Engineering, Yonsei University, Seoul 03722, South Korea

*Correspondence to*: Yeonjoo Kim (yeonjoo.kim@yonsei.ac.kr)

**Abstract.** Wildfires influence not only ecosystems but also carbon and water fluxes on Earth. Yet, the fire processes are still limitedly represented in land surface models (LSMs), thus simulating the occurrence and consequences of fires. Especially, the performance of LSMs in estimating burned areas across high northern latitudes is poor. In this study, we employed the
daily burned areas from the satellite-based global fire emission database (version 4) (GFED4) into the community land model (version 5.0) with a biogeochemistry module (CLM5-BGC) to identify the effects of accurate fire simulation on carbon and water fluxes over Alaska and Eastern Siberia. The results showed that the simulated carbon emissions with the burned areas from GFED4 (i.e., experimental run) were significantly improved in comparison to the open-loop run (i.e., default run), which resulted in opposite trends of the net ecosystem exchange for 2004, 2005, and 2009 over Alaska between the open-loop and
experimental runs. Also, we identified carbon emissions were more sensitive to the wildfires in Alaska than in Eastern Siberia, which could be explained by the vegetation distribution (i.e., tree cover ratio). In terms of water fluxes, canopy transpiration in Eastern Siberia was relatively insensitive to the size of burned area due to the interaction between leaf size and soil moisture. This study uses CLM5-BGC to improve our understanding of the role of burned areas in eco-hydrological processes at high latitudes. Furthermore, we suggest that the improved approach will be required for better predicting future carbon fluxes and
climate change.

## 1 Introduction

Wildfires are natural phenomena that directly and indirectly affect the life of humans as well as vegetated ecosystems (Bowman et al., 2009; Haque et al., 2021; Holloway et al., 2020; Li et al., 2017). Wildfires burn the leaves, stems, and roots of plants and alter ecological communities, which is called secondary succession (Knelman et al., 2015; Metrak et al., 2008; Seo and
Kim et al., 2019). Moreover, annual carbon emissions from wildfires were estimated to be approximately 2.1 Pg, which remarkably affects the global carbon cycle (Arora and Melton et al., 2018; van der Werf et al., 2010). Wildfires can be a potential disaster that result in enormous damage; for example, the damage costs of Australian wildfires from 2019 to 2020 was estimated to be over $100 billion, covering infrastructure damage, job losses, and firefighting cost (Deb et al., 2020). Moreover, the smoke particles from wildfires may be harmful to human health (Cascio, 2018; Black et al., 2017).





Especially, at high latitudes areas, such as boreal forest and tundra regions, the wildfire intensity and occurrence have increased over the past decades (Jiang et al., 2015; Madani et al., 2021; Veraverbeke et al., 2017). While few arctic fires had occurred historically because of the low temperatures in summer season, snow cover, and short growing seasons, arctic fires are no longer unusual owing to warming trends. For instance, unprecedented large fires (more than 1.5 Mha of burned areas) in

interior Alaska were reported in 2004 and 2015. From these fires, more than 50 Tg C was emitted, according to the Alaskan Fire Emissions Database (AKFED) (Veraverbeke et al., 2015). These fires not only result in carbon emissions from vegetation but also increase the soil temperature in summer, which could induce permafrost thawing (Holloway et al., 2020; Jiang et al., 2015). This could result in the release of carbon in belowground regions, which can increase the levels of carbon dioxide in the atmosphere.


Fires at high latitudes are mainly caused by natural processes rather than by humans. Veraverbeke et al. (2017) reported that 76–87% of fire ignition and 82–95% of burned areas were the result of the lightning occurring between 1975 and 2015 in North American boreal forests. They also suggested that persisting warming and dryness accelerate the spread of fires, which could cause extreme fires. Furthermore, their regression analysis showed that lightning frequency will increase in the future

(2050–2074), which may increase the burned area in Alaska. Therefore, understanding the fire mechanism is critical to predict future fires and carbon emissions as well as evaluate the fire risk to permafrost carbon.

To understand and describe wildfire dynamics, many fire models such as Community Land Model (CLM)-Li (Li et al., 2012), SPread and InTensity of FIRE (Thonicke et al., 2010), MC-Fire (Conklin et el., 2015), Fire Including Natural & Agricultural

Lands model (Rabin et el., 2016), and the interactive fire and emission algorithm for natural environments (Mangeon et al., 2016), which have been incorporated into earth system models (ESMs) and land surface models (LSMs), have been developed. As individual fire models were developed for different purposes, each model calculates fire ignition, burned area, fire combustion, and mortality based on different structures of fire regime and input data. The Fire Modeling Intercomparison Project (FireMIP; Rabin et el., 2017) was executed for comparing the performances of these fire models and assessing their

strengths and weaknesses in details. Despite these efforts of developing fire models, LSMs are still limited in representing the burned area thus simulating fire impacts on the land surface processes. This is because understanding of a process-based fire mechanism remains elusive and thus large uncertainties of fire parameterization exist (Wu et al., 2021).

In this study, we aimed to understand the significance of fire prediction in further simulating fire impacts on ecohydrological

processes in the LSMs. We implemented the daily burned areas derived from Global Fire Emissions Database (GFED) for twelve years (2001–2012) over the arctic region into the National Center for Atmospheric Research (NCAR) CLM version 5.0 with a biogeochemistry module (CLM5-BGC), one of the widely used LSMs. In CLM5-BGC, the burned area is predicted based on the empirical relationships among lightning frequency, human population density and vegetation composition, which


is limited in capturing the observed burned areas from the GFED over several areas, including those at high latitudes. We

compared the results of the open loop CLM5-BGC simulation (hereafter, OL, which uses the default fire module) and the experimental simulation with GFED4 (hereafter, EXP-GFED4) with a focus on Alaska and Siberia, where there are large uncertainties of fire prediction (i.e., prediction of burned area). Furthermore, we examined the simulated carbon fluxes and water fluxes, including evapotranspiration (ET) and soil moisture in OL and EXP-GFED4.

## 2 Model and data

### 2.1 CLM5-BGC

CLM5, a land component of the NCAR community earth system model (version 2.0.1), is a grid-based computational model (Lawrence et al., 2019). Each grid cell comprises of sub-grids that represent the land cover type (i.e., glacier, lake, wetland, urban, and vegetated). The 17 plant functional types (PFTs) are represented in the vegetated land cover. The model represents the instantaneous exchange of energy, and water and momentum were simulated between terrestrial and atmosphere across a

variety of spatial and temporal scales at the sub grid level. Furthermore, hydrological processes including evapotranspiration, surface runoff, sub-surface runoff, stream flow, aquifer recharge, and snow are simulated at the sub-grid level. When the BGC module is adopted (i.e., CLM5-BGC), the carbon and nitrogen cycles and seasonal vegetation phenology are simulated for the atmosphere, vegetation, and soil organic matter at the PFT-level. These cycles, which are linked to climate, land cover and land use, fires, and atmosphere $CO_2$ level, affect other cycles such as hydrological cycles and energy fluxes.


In CLM5-BGC, fire is simulated based on a process-based fire parameterization developed by Li et al. (2012). There are four types of fire in CLM5-BGC: non-peat fire, agriculture fire, deforestation fire, and peat fire. For non-peat fires, the number of fire ignitions is calculated as the sum of natural and anthropogenic ignitions. The estimation of natural ignition sources is based on the NASA lightning imaging sensor (LIS) / optical transient detector (OTD) lightning frequency datasets. The frequency

of cloud-to-ground lightning that ignite fires is estimated with the latitudinally varying ratios of total lightning frequency obtained from remotely sensed data (i.e., LIS/OTD), which include two different types of lightning, i.e., cloud-to-ground and the cloud-to-cloud lightning. Furthermore, the ignition source from human activity is calculated based on the human population density. The fire spread rate is then calculated by considering wind speed and vegetation condition (Arora and Boer, 2005). Socioeconomic influences are parameterized using GDP and population density, which means that higher populated and more

developed regions will have a better fire suppression capacity.

The PFT-level carbon emissions from fires are calculated as follows:

$$CE = A \cdot C \cdot CC \qquad (1)$$




where $CE$ is carbon emission at the PFT level; $A$ is the burned area at the PFT level; $C$ is a vector with the carbon density of leaves, stems, and roots, carbon transfer, and carbon pools; and $CC$ is the corresponding combustion completeness factor vector.

Leaves and roots may be damaged in burned areas, which reduces their carbon-capturing productivities (Reyer et al., 2017; Seo and Kim, 2019; Swezy and Agee, 1991). In CLM5-BGC, the amount of leaf carbon to litter ($\Psi$) caused by fire is calculated as follows:

$$\Psi = \frac{A_b}{f_i \cdot A_g} \cdot C_{leaf}(1 - CC) \cdot M \qquad (2)$$

where $A_b$ is the calculated burned area, $A_g$ is the area of the grid cell, $f_i$ is the fraction of coverage of each PFT, $C_{leaf}$ is the amount of leaf carbon, and $M$ is the mortality factor vector for each PFT. The leaf size is recalculated based on the adjusted amount of leaf carbon. In addition, the methods by which the amount of carbon in live stems, dead stems, and roots and the storage pool are adjusted due to fires are similar to those mentioned above.

Leaf size controls canopy evaporation and transpiration as well as carbon fluxes (gross primary production (GPP), net primary production (NPP), net ecosystem production (NEP), and net ecosystem exchange (NEE)). Especially, the NEE, which represent the total carbon fluxes between an ecosystem and the atmosphere, is calculated by using the NEP and carbon emissions from wildfires. The equations for these carbon fluxes are as follows:

$$NPP = GPP - R_p \qquad (3)$$

$$NEP = NPP - R_h \qquad (4)$$

$$NEE = -NEP + CE \qquad (5)$$

where $R_p$ is plant respiration and $R_h$ is heterotrophic respiration.

Because hydrological processes are highly linked to vegetation dynamics, fire processes may affect not only water cycles but also ecosystem products (Jiao et al., 2017). For instance, the water cycles on land surfaces, such as partitioning of ET, are affected by fires because the fire changes leaf size in ecosystems (Netzer et al., 2009; Park et al., 2020; Seo and Kim, 2019; Wang et al., 2019). More details on CLM5-BGC processes, including the equations for leaf phenology, hydrology cycles, fires, and carbon cycles, are described in Lawrence et al. (2019).





### 2.2 Fire and carbon fluxes datasets

The GFED (version 4), which is based on satellite data such as MODIS and Tropical Rainfall Measuring Mission Visible and Infrared Scanner, provides gridded data on global burned area, fire persistence, land cover distribution, and fractional tree cover distribution of burned areas, among others (Giglio et al., 2013). The data are provided at a 0.25° × 0.25° resolution and daily and monthly temporal resolutions. Furthermore, details on fire impacts, such as carbon emissions, dry matter emissions, biosphere fluxes (NPP, heterotrophic respiration) and emission factors data are included. The carbon emission data are based on burned areas and the Carnegie-Ames-Stanford Approach (CASA) carbon-cycle terrestrial model for each month. In this study, daily burned area data from GFED4 were incorporated into CLM5-BGC, and monthly scaled carbon emission data from GFED4 were used to evaluate the model performance (Table 1).

We also used data on Alaskan carbon emissions from the AKFED to evaluate the model performance for carbon emissions in Alaska (Table 1). Veraverbeke et al. (2015) developed a statistical model to calculate the carbon consumption in Alaska between 2001 and 2012. They employed environmental variables such as elevation, slope, and day of burning to calculate ground-level carbon consumption. In addition, pre-fire tree cover and differenced normalized burn ratio are used to predict above ground carbon emission. They presumed that that the highest carbon emission was 69 Tg C in 2004 and the annual carbon emission was 15 Tg C.

We used monthly NEE products from GEOS-Carb CASA-GFED for 2003 to 2012 to evaluate the performance of EXP-GFED4. However, the definitions of NEE according to CLM5-BGC and GEOS-Carb CASA-GFED are quite different. In CLM5-BGC, the NEE is the final carbon flux between an ecosystem and the atmosphere. Thus, the carbon flux of burning was included when calculating the NEE (Eq. 5), but it was excluded in GEOS-Carb CASA-GFED. To unify the definition of NEE, we redefined the NEE in GEOS-Carb CASA-GFED as follows.

$$NEE = NEE_{ge} + FireE + FuelE \qquad (6)$$

where $NEE$ is the total carbon flux between terrestrial and atmosphere including emission dues to fires, $NEE_{ge}$ is the value of NEE according to GEOS-Carb CASA-GFED, $FireE$ is wildfire carbon emissions, and $FuelE$ is carbon emissions from wood-fuel burning.



## 3 Experimental design

### 3.1 Site description

In this study, we focused on Alaska (128E–139E, 45N–48N) and Eastern Siberia (128E–139E, 45N–48N), which are located at northern high latitudes (Figure 1). The average temperature based on CRU-NCEP reanalysis data (2001–2012) is -5.11 and -15.28 degrees Celsius in Alaska and Eastern Siberia, respectively. The average annual snowfall and rainfall are 83 mm and 218 mm in Alaska and 92 mm and 208 mm in Eastern Siberia, respectively.

There are differences in vegetation types in these regions, based on MODIS (Sun et al., 2008) (evergreen trees: 26.4%, deciduous trees: 1.6%, shrub: 28.5%, grass: 34.5%, crop: 3.9%, and bare ground: 5.1% in Alaska: evergreen trees: 1.2%, deciduous trees: 14.9%, shrub: 45.8%, grass: 29.7%, crop: 1.4%, and bare ground: 7.1 % in Eastern Siberia). In summary, the tree fraction is higher in Alaska (28%) than that in Eastern Siberia (16.1%), but the fraction of low vegetation (i.e., grasses and shrubs) is lower in Alaska (63%) than that in Eastern Siberia (74.5%).

### 3.2 Experimental design

In this study, we designed two sets of experiments to investigate the impact of burned area using fire simulation based on the study by Li et al. (2012) (i.e., OL) and satellite observations from GFED4 (i.e., EXP-GFED) to investigate the fire impact on the terrestrial model for Alaska and Eastern Siberia. OL and EXP-GFED4 were simulated at a spatial resolution of 0.5 longitude and 0.5 latitude using Climate Research Unit (CRU) – National Centers for Environmental Prediction (NCEP) reanalysis climate data, which include precipitation, temperature, wind speed, surface pressure, specific humidity, longwave radiation, and solar radiation. As shown in Figure 2, the spin-up simulation, which stabilizes the land state such as LAI, soil moisture, and soil temperature, was repeatedly run for 200 years using 20-year CRU/NCEP forcing data for 1980–2000 before adopting OL and EXP-GFED4. After the spin-up process, the two simulations were performed using 12 years (2001–2012) of CRU-NCEP forcings. While burned areas were simulated based on Li et al. (2012) in OL, the GFED daily burned area over the arctic region was directly inserted into CLM5-BGC in EXP-GFED4, with the daily data being equally divided into a half-hourly model timestep (Seo and Kim, 2022).

In this study, we compared the carbon and water fluxes in OL and EXP-GFED4. Especially, carbon emissions and the NEE were evaluated using the GFED database, AKFED, and GEOS-Carb CASA-GFED. Additionally, we analyzed the impacts of fire on carbon fluxes according to the distribution of PFT. Furthermore, comparisons of water fluxes such as ground evaporation, canopy evaporation, canopy transpiration, and soil moisture at grid level were performed to reveal the impacts of fire on water cycles.



## 4 Results and discussions

### 4.1 Burned area

We first evaluated the performance of estimating burned areas in Alaska and Eastern Siberia from CLM5-BGC (i.e., OL) and compared it to that of GFED4 (Figures 3). While an average of 0.42 Mha of burned area from 2001 to 2012 was observed in Alaska, the average of annual burned area was estimated at 0.24 Mha in OL (Figures 3a). In Alaska, there were large discrepancies of burned areas for 2004, 2005, and 2008 between the GFED4 and simulation results. More than 1 Mha of burned area existed for three years (2004, 2005, and 2009), which is remarkably different from that of the other years. Studies suggested that these big fires were associated with a high lightning frequency and drought (Littell et al., 2016; Veraverbeke et al., 2017; Xiao and Zhuang, 2007;). However, this phenomenon was not captured in CLM5-BGC, which predicts relatively constant annual burned areas. In contrast, the burned area was dramatically overestimated in Eastern Siberia (Figure 3b). While an average of 0.29 ha of burned area was observed, the average of annual burned area was estimated at 2.14 ha with CLM5-BGC. Especially, more than 4 Mha of burned area was simulated in 2011 and 2012 using CLM5-BGC.

Figure 4 shows an inadequately simulated spatial distribution of burned area over Alaska for 2004. The number of grid cells observed, with more than 0.1 ha of burned areas in 2004, was more than 50. In contrast to GFED4 burned areas, there were a few cells with more than 0.1 ha of burned areas simulated using CLM5-BGC in Alaska (Table 2). Table 2 shows that CLM5-BGC has weakness of simulating largely burned areas in Alaska. Small fires were simulated on more grid cells, and the simulated burned areas were more widely distributed than that in the GFED4 products.

These differences between the model and observation may be attributed to incorrect input data such as lightning frequency and fire management as well as a misunderstanding of fire processes. The processes of fire ignition and the fire spread are quite complex. Although the detailed position of lightning, weather condition, vegetation type, and vegetation complexity are considered to simulate fire processes more accurately, there is still the limitation of using point data in the grid-based model.

Furthermore, the limitation of fire ignition source may bring up the discrepancies between the modeled burned area and observed burned area. Lightning, which is a major source of fire at high latitudes, especially Alaska, has increased because of warming climate (Kępski and Kubicki, 2022). While the lightning frequency at high latitudes varied yearly, the climatology of 3-hourly lightning frequency from 1995 to 2011 was used in the CLM. Moreover, the calculated ratio of cloud-to-ground lightning has large uncertainties and may cause the models to misestimate fire ignition and burned areas.

The management system and infrastructures for fires varies by country or region. For instance, there are four steps of fire policy in Alaska, namely Full, Critical, Modified, and Limited, according to the levels of anthrophonic effort of extinguishing the fire (Phillips et al, 2022). Especially, there are no human impacts of fire suppression in Limited regions in Alaska. In




CLM5-BGC, however, the suppression impact is calculated based on the GDP and population, which could underestimate burned areas in Limited regions of Alaska because of the large GDP of the United States.

## 4.2 Fire impacts on carbon fluxes

We compared the carbon fluxes of OL and EXP-GFED4 to understand the impacts of fire on high latitudes regions (Fig. 5 and 6 and Table 3). The average carbon emissions were 11.87 and 21.11 g m$^{-2}$ year$^{-1}$ in OL and EXP-GFED4 in Alaska, respectively, and 20.48 and 3.24 g m$^{-2}$ year$^{-1}$ in OL and EXP-GFED4 in Eastern Siberia, respectively (Table 3). As expected, there were large differences in carbon emissions in OL and EXP-GFED4 in both regions because the simulated carbon emission was directly linked to burned areas. In the model, carbon emissions had a strong correlation with burned areas in both regions (Alaska: 0.99, Eastern Siberia: 0.89).

Furthermore, the simulated Alaskan annual carbon emission for OL and EXP-GFED4 were evaluated with AKFED carbon emission datasets and GFED4 (Fig. 5a and Table 4). The correlations of annual carbon emission between simulated carbon emissions (OL and EXP-GFED4) and GFED4 were 0.3 and 0.99, respectively. Moreover, the correlation was further determined by comparing the results with the AKFED carbon emissions (OL: 0.31 and EXP-GFED4: 0.96). While the root mean square error (RMSE) of the simulated carbon emissions decreased with applying the GFED4 burned area, comparing the AKFED products (OL : 20.48 g m$^{-2}$ year$^{-1}$ g/m2/year and EXP-GFED4 : 10.98 g m$^{-2}$ year$^{-1}$g/m2/year), the it increased comparing the GFED4 carbon emissions (OL: 11.02 g m$^{-2}$ year$^{-1}$ g/m2/year and EXP-GFED4: 20.93 g m$^{-2}$ year$^{-1}$g/m2/year). This is because average carbon emissions for GFED4 were 8.36 g m$^{-2}$ year$^{-1}$ and are relatively lower than carbon emissions in EXP-GFED4 and AKFED, which is consistent with Veraverbeke et al. (2015), showing that the carbon emissions from the AKFED were higher than those for GFED3s. They assumed that there was a possibility that GFED3s underestimated carbon combustion due to fires in boreal forest regions. The simulated carbon emissions of EXP-GFED4 were generally higher than those of GFED4 and AKFED. Therefore, the corresponding combustion completeness factor vector for boreal trees may be lower than the current default value in CLM5-BGC.

The carbon emission simulation was highly improved after replacing the fire simulation with GFED4 in Eastern Siberia (Fig. 5b); the correlation was improved from 0.41 in OL to 0.88 in EXP-GFED4 and the RMSE was reduced form 19.74 g m$^{-2}$ year$^{-1}$ in OL to 4.2 g m$^{-2}$ year$^{-1}$ in EXP-GFED4 with comparing GFED4 products. In Eastern Siberia, grasses are dominant, suggesting that the value of the corresponding combustion completeness factor vector for grass is more reliable than that for boreal trees (dominant in Alaska) in CLM5-BGC.

The simulated LAIs in Alaska and Eastern Siberia are presented in Fig. 6a and 6b, respectively. In Alaska (Fig. 6a), the difference in LAI between OL and EXP-GFED4 was the largest in 2005 (0.029 m$^2$/m$^2$). Although the difference in burned area between OL and GFED4 (Fig. 3a) was the largest in 2004, the largest difference in LAI was in 2005 since vegetation damage



caused by fire in 2004 had not fully recovered, and the difference in burned area in 2005 was also quite large. In Eastern Siberia (Fig. 6b), the difference in the simulated LAI between OL and EXP-GFED4 has been large since 2009, when the difference in size of burned areas was amplified (Fig. 3b).

Unlike carbon emissions, regionally-averaged GPP, NPP, and NEP (Fig. 6c–6h) did not significantly change in EXP-GFED4. The rates of change in GPP, NPP, and NEP are less than 3%, indicating that fires rarely impacted carbon fluxes related to vegetation and decomposition. This is because the ratio of the fire area to the total area was relatively small. For example, the highest annual burned area of all simulations was 6 Mha, which accounted for 6.87% of our study domain. Although the LAI, which affects primary GPP and other carbon fluxes, was reduced by fires, the LAI after fires was not substantially different

due to the small fire area compared to the total area.

However, NEE, which represents the net carbon fluxes between terrestrial and ecosystem (Eq. 5), was largely affected by fires, unlike other fluxes such as GPP, NEE, and NEP (Fig. 5c and 5d). NEE changed significantly with forcing of GFED4 into the model when the discrepancy of burned area between OL and EXP-GFED4 was remarkable. Moreover, the NEE results for

EXP-GFED4 and GEOS-Carb CASA-GFED had similar tendencies. For instance, we found that the net carbon in Alaska was emitted from land ecosystems to the atmosphere (i.e., positive NEE) in 2004, 2005, and 2009 in EXP-GFED4 and GEOS-Carb CASA-GFED, but it was absorbed (i.e., negative NEE) in OL. Although there was a change in NEE due to burned areas in Siberia, it was not as pronounced as that in Alaska.

Therefore, one can tell that the carbon fluxes were more sensitive in Alaska than Eastern Siberia. The reasons for carbon emissions being more pronounced in Alaska than Eastern Siberia could be explained by the vegetation distribution. The average ratio of carbon emission to burned area was 49.98 Tg Mha$^{-1}$ in Alaska and 9.76 Tg Mha$^{-1}$ in Eastern Siberia. There was 95 Tg of leaf carbon and 8.3 Tg of live-stem carbon in Alaska and 29 Tg of leaf carbon and 2.4 Tg of live-stem carbon in Eastern Siberia in averages of OL and EXP-GFED4. Trees have a larger LAI and stem and thus more fuel combustibility and

availability. Therefore, the ratio of carbon emission to burned area was high in forest than in grassland. Therefore, the final carbon fluxes between the atmosphere and vegetation were closely linked with not only vegetation metabolism but also burned area and plant type.

By comparing our experiments of OL and EXP-GFED4 in Alaska and Eastern Siberia, we understand the impact of fires on

carbon fluxes. CLM5-BGC is still limited in representing fire processes and, thus, in simulating the occurrence and consequences of fires. Therefore, there is a large difference in burned area between simulation and observation. The results indicated that the application of satellite-based observations of burned areas remarkably improved carbon emission estimations, while showing that opposite NEE trends were simulated between OL and EXP-GFED4 in Alaska. Moreover, we determined





the vegetation distribution in burned areas, which suggests a link between carbon emission sensitivity to fire and total carbon
fluxes.

### 4.3 Carbon fluxes in grid level

The results of the carbon fluxes at the grid level in Alaska and Eastern Siberia are investigated in Fig. 7, which shows the difference of carbon fluxes and burned areas between OL and GFED4 in Alaska for 2004 and in Eastern Siberia for 2012. As expected, the response of GPP, NPP, and NEP to fires were nonsignificant. However, fires significantly altered carbon
emissions and the NEE in both regions, which can further alter the atmospheric carbon dioxide concentration and even climate. This suggests that high-latitudes fires may influence the carbon sink or uptake markedly. Phillips et al. (2022) reported that boreal forest fires, which are largely distributed at high latitudes, make a significant contribution to releasing greenhouse gases. With the earth system model combined with CLM5-BGC, the prediction of atmospheric carbon may become uncertain due to the limited performance of fire prediction models.


Figure 8 shows the responses of carbon flux to changes in the burned area. The average change rates (difference in carbon fluxes/difference in burned area) of GPP, NEP, and NPP were -97.69, -5.12, and -32.27 gC ha$^{-1}$ and -55.72, 32.83 and 26.55 gC ha$^{-1}$ in Alaska and Eastern Siberia, respectively. The NPP was slightly positively correlated with fires because plant respiration is more sensitive compared to GPP in Eastern Siberia. In other words, if the burned area increases, both GPP and
plant respiration will decrease. As plant respiration decrease more than GPP, it was simulated that NPP increases with the frequency of fires in Eastern Siberia with CLM-BGC5.

The average change rates of NEE and carbon fluxes were 4913.7 and 4881.4 gC ha$^{-1}$ and 771.02 and 797.58 gC ha$^{-1}$ in Alaska and Eastern Siberia, respectively. The response of carbon emissions to fires was much more sensitive than those of GPP, NPP,
and NEP; thus, changes in carbon emissions are a major cause of the change in the NEE, which is consistent with previous results. Carbon release due to the wildfires was more sensitive in Alaska than Eastern Siberia under CLM5-BGC, as boreal trees are more distributed in Alaska than Eastern Siberia. Based on the above results, we suggest that more accurate fire predictions are needed to understand ecosystem carbon fluxes, especially in Alaska.

### 4.3 Fire impacts on water fluxes

To investigate the fire impacts on water fluxes, we compared the results of ET and ET components such as canopy evaporation, canopy transpiration, and ground evaporation in six grids where the differences in burned area between OL and EXP-GFED4 are largest in Alaska and Eastern Siberia (Fig. 9). Because the LAI was affected by wildfires, canopy evaporation and canopy transpiration decrease in the burned areas. Moreover, we may find that more rainfall reaches the ground, which would make the ground evaporation rate higher in regions with more burned areas, especially in 2004 and 2005 in Alaska. The differences
in annual canopy evaporation, canopy transpiration, and ground evaporation were 5.41 mm and 13.37 mm, 2.3 mm and 6.26





mm, and -1.39 mm and -7.4 mm in 2004 and 2005, respectively. This is consistent with Li et al. (2017) and Seo and Kim (2019), showing the canopy evaporation and canopy transpiration would decrease when comparing the simulation with and without fire. Furthermore, Seo and Kim (2019) compared the CLM-dynamic global vegetation model with and without fire and found that fire decreased canopy evaporation and canopy transpiration but increased ground evaporation. Therefore, the
total ET in the presence of fire decreased by 6.32 mm and 12.08 mm in 2004 and 2005, respectively.

In Eastern Siberia, the patterns of canopy evaporation and ground evaporation were the same as those of Alaska. Canopy evaporation increased and ground evaporation decreased in EXP-GFED4 because the simulated burned area decreased, which was noticeable from 2009 to 2012 (Fig. 9f and 9h). However, the canopy transpiration of EXP-GFED4 was similar with that
of OL. In other words, there was no significant change in canopy transpiration due to change in burned area. Furthermore, the ET with the burned area applied changed slightly in Eastern Siberia. Differences in the average canopy evaporation and ground evaporation were -9.19 mm and 6.97 mm from 2009 to 2012, respectively. The reasons for the smaller change in canopy transpiration is related to soil moisture and leaf size.

Figure 10 shows differences in the simulated soil moisture for OL and EXP-GFED4 at 0–20 cm (hereafter top soil) and 70–150 cm (hereafter bottom soil) in both regions. In Eastern Siberia, the top soil moisture and bottom soil moisture decreased after applying the observed burned areas. Although the leaf size increased with less burned areas applied, transpiration did not change significantly due to the decreased soil moisture. On the contrary, there was no considerable difference in the top and bottom soil moisture between OL and EXP-GFED4. Therefore, transpiration was positively correlated with leaf size.
According to the McVicar et al. (2012) and Nemani et al (2003), the Alaska region is drier and more water-limited than Eastern Siberia. Energy is sufficient to evaporate the increased stored water from the ground, which explains why soil moisture did not change considerably in Alaska.

## 5 Conclusions

In this study, we applied the daily burned area of GFED4 into CLM5-BGC over Alaska and Eastern Siberia. As the capacity
of predicting the burned area with CLM5-BGC in high latitudes is poor, the simulated burned area was overestimated in Eastern Siberia, and it was underestimated in Alaska. Such model discrepancy could lead to misunderstanding of terrestrial carbon and water fluxes. While GPP, NPP, and NEP were not significantly affected by burned area, carbon emissions changed considerably in both regions; thus, NEE was significantly influenced by the burned area. Furthermore, carbon emissions were remarkably improved after applying GFED4 into CLM5-BGC, which caused opposite trends of simulated NEE between the
OL and EXP-GFED4 for 2004, 2005, and 2009 in Alaska. In addition, the density of leaf and stem carbon in Alaska were much higher than those in Siberia, indicating that carbon emissions from fire in Alaska are more sensitive than those in Siberia.



Furthermore, while analysis of burned area impact on water fluxes showed that canopy evaporation and ground evaporation were changed consistently by fires, canopy transpiration and soil moisture were affected by the region. For example, canopy
transpiration in Eastern Siberia was almost the same for OL and EXP-GFED4, because the leaf size was larger and soil moisture decreased due to reduced fires. However, the transpiration of EXP-GFED4 decreased as the leaf size was smaller but there was no significant change in soil moisture in Alaska. This may have been because Alaska is a more water-limited region; thus, energy is sufficient to evaporate the increased stored water from the ground. Although an accurate estimation of carbon cycles is necessary to predict the future climate, we found that the fire model was limited in representing burned areas and, thus, in
simulating carbon emissions and the NEE. Therefore, we suggest that innovative methods for simulating burned areas (i.e., using machine learning) should be required to better predict future carbon fluxes and climate change.

**Code and Data Availability**

CLM5, a land part of CESM 2.0.1, is available on GitHub at https://github.com/escomp/cesm.git (last access: 20 December
2022). GFED4 products are available at https://daac.ornl.gov/VEGETATION/guides/fire_emissions_v4_R1.html. Carbon emissions database from AKFED and was available at https://daac.ornl.gov/CARVE/guides/AKFED_V1.html. NEE products from          GEOS-Carb          CASA-GFED          were          available          at https://disc.gsfc.nasa.gov/datasets/GEOS_CASAGFED_3H_NEE_3/summary. The revised codes, which enable the application of GFED4 into CLM5-BGC, are achieved on Zenodo at https://zenodo.org/record/7483115 (Seo and Kim, 2022).

**Author contributions**

HS and YK designed the study, and HS performed the model development, simulations, and result analysis under the supervision of YK. HS wrote the original manuscript, and YK reviewed and edited the manuscript.

**Competing interests**

The authors declare that they have no conflict of interest.

**Acknowledgements**

This study was supported by the Korea Polar Research Institute (KOPRI) funded by the Ministry of Oceans and Fisheries (PE22900) and the Basic Science Research Program through the National Research Foundation of Korea, which was funded by the Ministry of Science, ICT & Future Planning (2020R1A2C2007670).



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




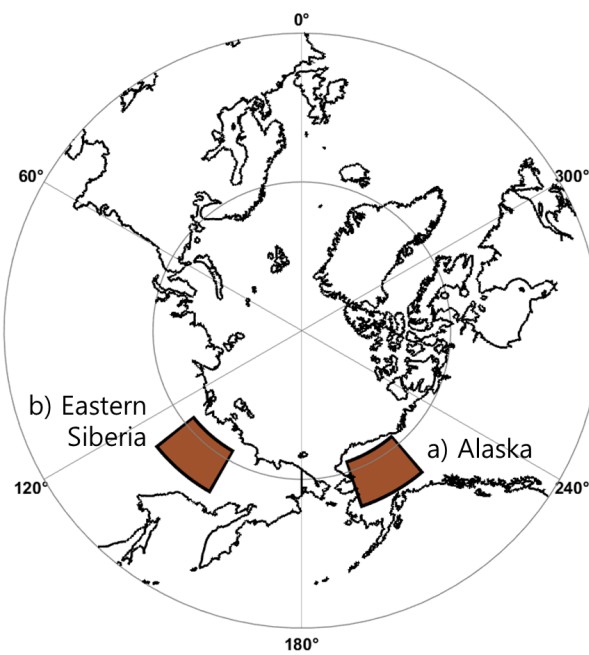

**Figure 1. Study domain, located between 128° E–139° E and 45° N–48° N (a) Alaska, and (b) Eastern Siberia, located between 128° E–139° E and 45° N–48° N.**



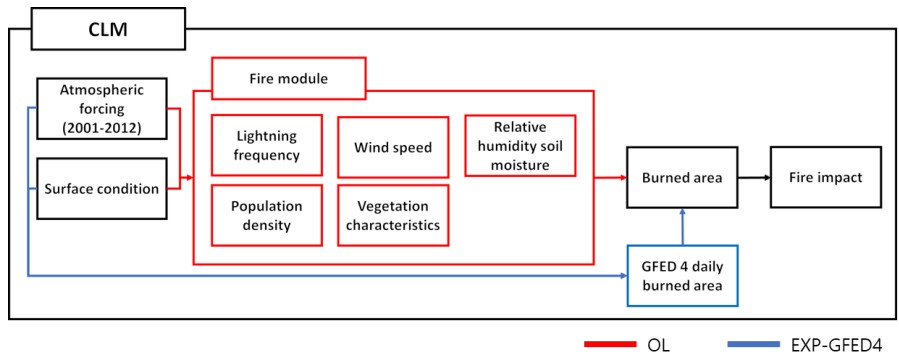


**Figure 2. Flow diagram for OL (red line) and EXP-GFED4 (blue line).**

OL, open loop CLM5-BGC simulation; EXP-GFED4, experimental simulation with global fire emission database



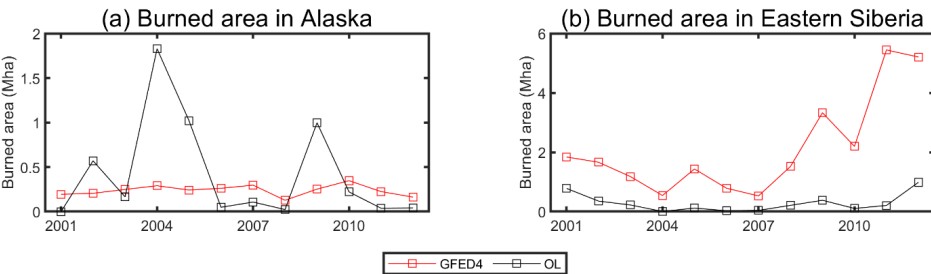

**Figure 3. Burned area based on GFED4 and simulated burned area of OL over (a) Alaska (b) and Eastern Siberia from 2001 to 2012.**

GFED4, global fire emission database (version 4); OL, open loop CLM5-BGC simulation




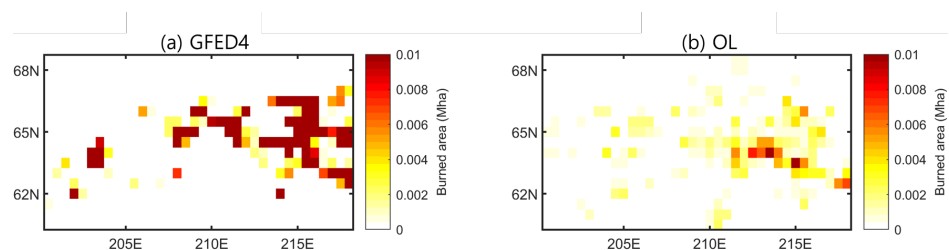

**Figure 4. Spatial distribution of burned area of (a) GFED4 (b) and OL in 2004 over Alaska.**

GFED4, global fire emission database (version 4); OL, open loop CLM5-BGC simulation





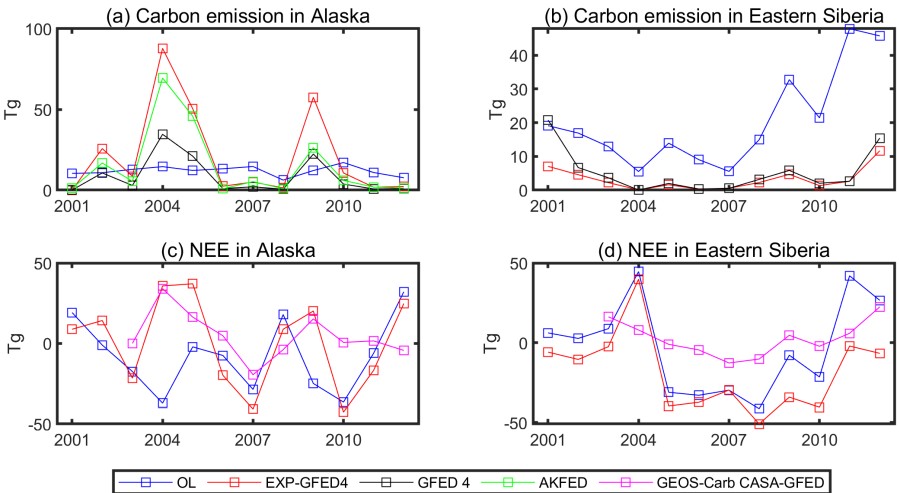

**Figure 5. Simulated carbon fluxes of OL (blue line) and EXP-GFED4 (red line) such as carbon emission (a,b), and (c,d) in Alaska (a,c) and Eastern Siberia (b,d) from 2001 to 2012. GFED carbon emission (a,b, black line) and AKFED carbon emission (a, green line) are added to show the impact of burned assimilation to carbon emission. Also, NEE of GEOS-Carb CASA-GFED was added to evaluate the performance of NEE in OL and EXP-GFED4 runs (c, d, magenta line).**

OL, open loop CLM5-BGC simulation; EXP- GFED4, experimental simulation with global fire emission database (version 4); AKFED; Alaskan Fire Emissions Database; NEE, net ecosystem exchange



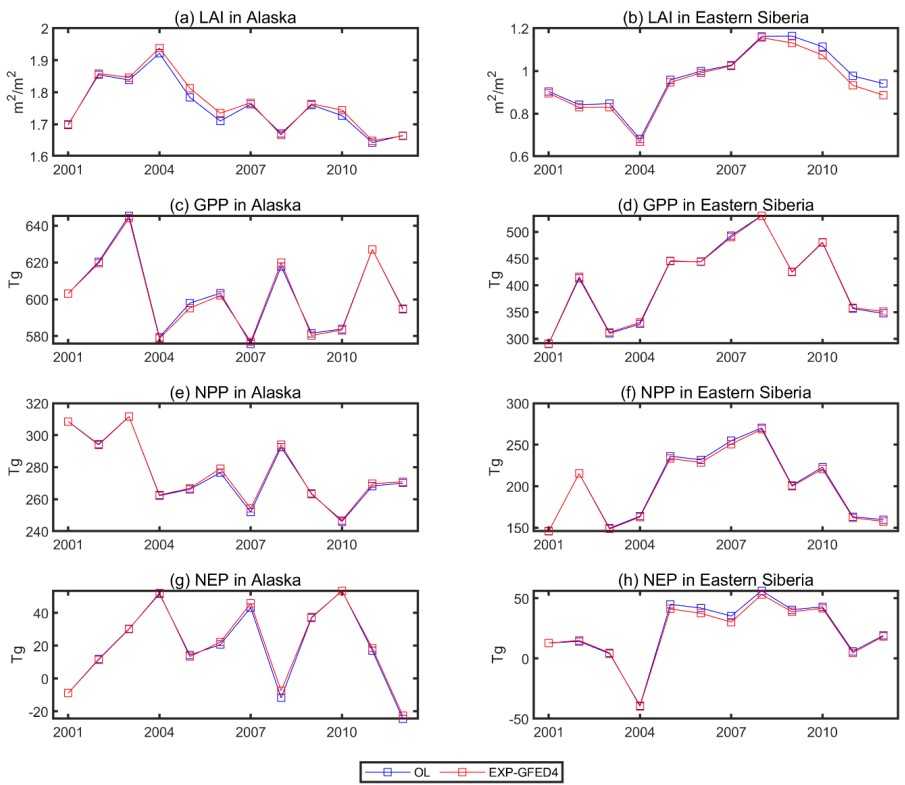


**Figure 6. Simulated LAI (a,b) and carbon fluxes of OL (blue line) and EXP-GFED4 (red line) such as GPP (c,d), NPP (e,f), and NEP (g,h) in Alaska (a,c,e,g) and Eastern Siberia (b,d,f,h) from 2001 to 2012.**

LAI, leaf area index; OL, open loop CLM5-BGC simulation; EXP-GFED4, experimental simulation with global fire emission database (version 4); GPP, gross primary production; NPP, net primary production; NEP, net ecosystem production


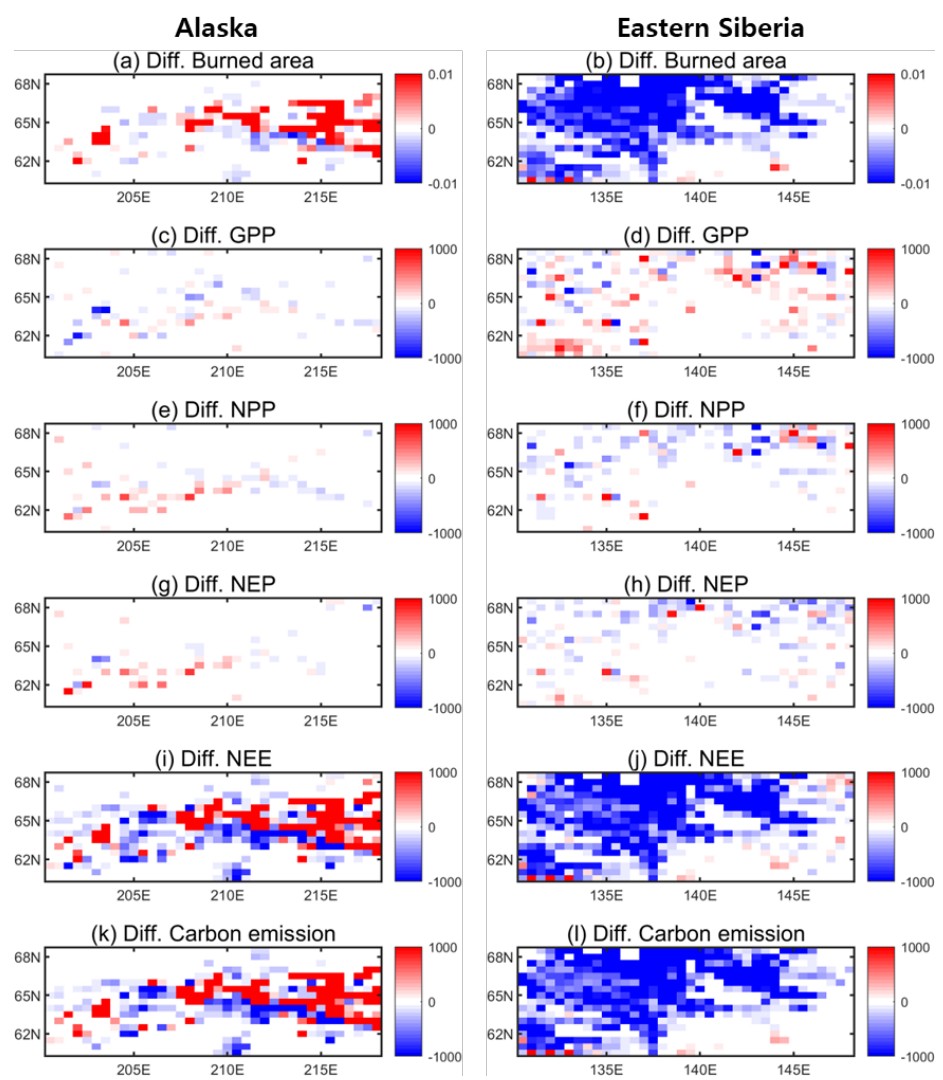

**Figure 7. Map of difference of burned area (a,b) and carbon fluxes such as GPP (c,d), NPP (e,f), NEP (g,h), NEE (i,j), and carbon emission (k,l) in 2004 over Alaska (a,c,e,g,i,k) and in 2012 over Eastern Siberia (b,d,f,h,j,l).**

GPP, gross primary production; NPP, net primary production; NEE, net ecosystem production



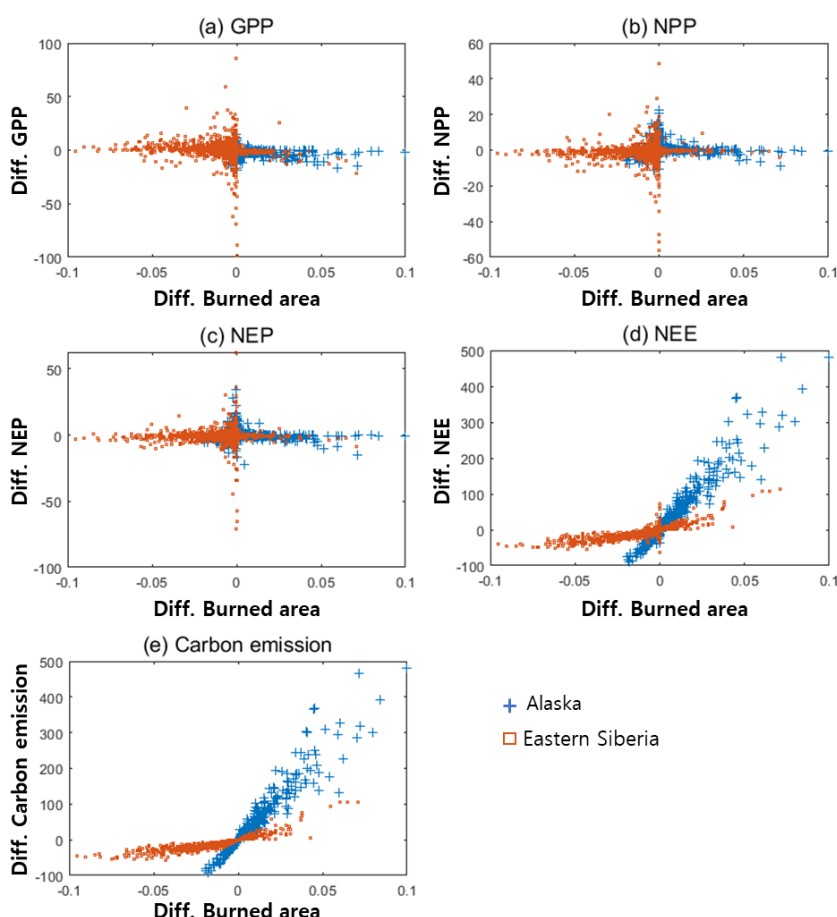

**Figure 8. The responses of the GPP (a), NPP (b), NEP (c), NEE (d), and carbon emission (e) to burned area over Alaska and Eastern Siberia.**

GPP, gross primary production; NPP, net primary production; NEP, net ecosystem production; NEE, net ecosystem production



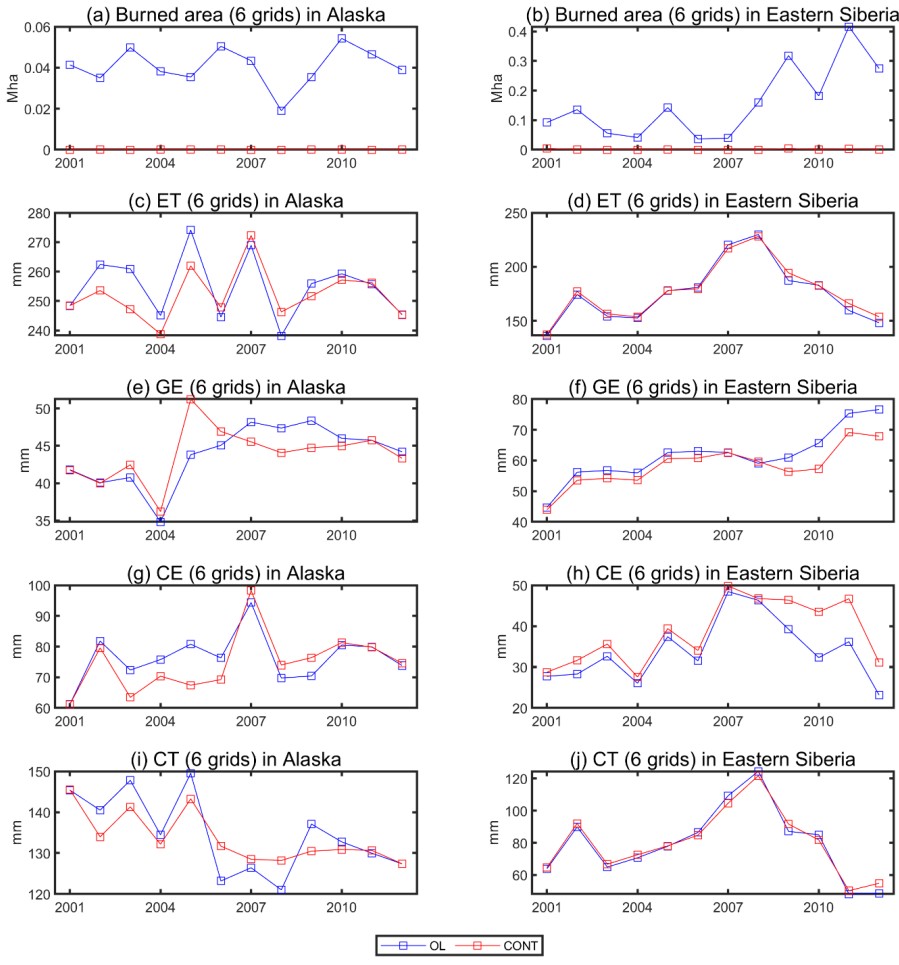

**Figure 9. Simulated burned area (a,b) and water fluxes of OL (blue line) and EXP-GFED4 (red line) such as ET (c,d), ground evaporation (GE, e,f), canopy evaporation (CE, g,h) and canopy transpiration (CT, i,j) in 5 grids where the difference in burned area between OL and EXP-GFED4 is highest in Alaska (a,c,e,g,i) and Eastern Siberia (b,d,f,h,j) from 2001 to 2012.**

OL, open loop CLM5-BGC simulation; EXP-GFED4, experimental simulation with global fire emission database (version 4); ET, evapotranspiration

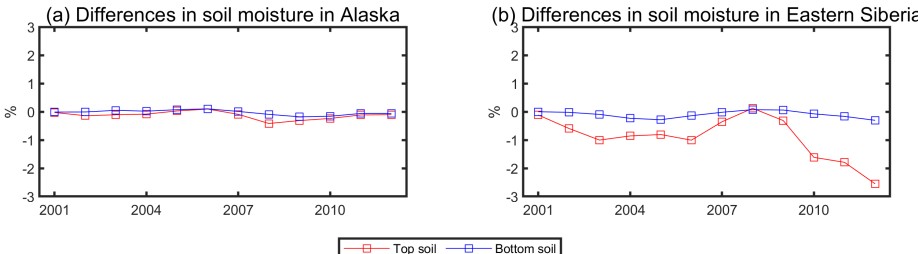

**Figure 10. Differences (the value of OL- the value of EXP-GFED4) in simulated top soil (0–20 cm) moisture and bottom soil (70–150 cm) moisture in Alaska (a) and Eastern Siberia (b).**

OL, open loop CLM5-BGC simulation; EXP-GFED4, experimental simulation with global fire emission database (version 4)





**Table 1. Model and data in this study.**

| Model | Domain and simulation period | Reference |
|---|---|---|
| Community Land Model 5 - Biogeochemistry | Alaska and Eastern Siberia (2001-2012) | Lawrence et al. (2019) |
| Data | Source | Reference |
| Burned area | GFED4 | Giglio et al. (2013) |
| Carbon emission | GFED4 | Giglio et al. (2013) |
| | AKFED | Veraverbeke et al. (2015) |
| NEE | GEOS-Carb CASA-GFED | Ott (2020) |

GFED4, global fire emission database (version 4); NEE, net ecosystem exchange; AKFED, Alaskan Fire Emissions Database; GEOS-Carb CASA-GFED,





**Table 2. Number of grid cells with more than 0.1 ha of burned area of GFED4 and OL.**

| Year | Number of grid cells (> 0.1Mha) | |
|------|------|------|
| | GFED v4 | OL |
| 2001 | 0 | 2 |
| 2002 | 21 | 1 |
| 2003 | 8 | 3 |
| 2004 | 51 | 2 |
| 2005 | 38 | 1 |
| 2006 | 2 | 3 |
| 2007 | 3 | 2 |
| 2008 | 0 | 0 |
| 2009 | 31 | 1 |
| 2010 | 7 | 3 |
| 2011 | 0 | 3 |
| 2012 | 1 | 2 |

GFED4, global fire emission database (version 4); OL, open loop CLM5-BGC simulation



**Table 3. Simulated carbon fluxes; carbon emission, GPP, NPP, NEP, and NEE in CLM-Default and CLM-GFED over Alaska and Eastern Siberia.**

| Units (gC/m²/year) | Alaska | | Eastern Siberia | |
|---|---|---|---|---|
| | OL | EXP-GFED4 | OL | EXP-GFED4 |
| Carbon emission | 11.87 | 21.12 | 20.48 | 3.24 |
| GPP | 602.51 | 602.12 | 405.16 | 406.14 |
| NPP | 276 | 276.79 | 201.2 | 199.49 |
| NEP | 19.5 | 20.42 | 23.28 | 21.56 |
| NEE | -7.63 | 0.7 | -2.79 | -18.32 |

GPP, gross primary production; NPP, net primary production; NEP, net ecosystem production; NEE, net ecosystem exchange; OL, open loop CLM5-BGC simulation; EXP-GFED4, experimental simulation with global fire emission database






**Table 4. Carbon emission of OL, EXP-GFED4, GFED4 and AKFED from 2001 to 2012 over Alaska.**

| Carbon emission (g/m2/year) | OL | EXP-GFED4 | GFED 4 | AKFED |
|---|---|---|---|---|
| 2001 | 10.37 | 0.05 | 0.04 | 1.16 |
| 2002 | 10.77 | 25.71 | 10.63 | 16.76 |
| 2003 | 12.85 | 8.59 | 2.88 | 5.48 |
| 2004 | 14.53 | 87.81 | 34.56 | 69.43 |
| 2005 | 12.18 | 50.43 | 21.02 | 45.78 |
| 2006 | 13.22 | 2.44 | 0.97 | 0.82 |
| 2007 | 14.62 | 4.97 | 2.03 | 5.26 |
| 2008 | 6.20 | 1.38 | 0.54 | 0.87 |
| 2009 | 12.10 | 57.49 | 22.32 | 26.30 |
| 2010 | 17.09 | 10.62 | 3.74 | 6.02 |
| 2011 | 10.89 | 1.79 | 0.72 | 1.86 |
| 2012 | 7.64 | 2.19 | 0.88 | 1.21 |
| Average | 11.87 | 21.12 | 8.36 | 15.08 |

OL, open loop CLM5-BGC simulation; EXP-GFED4, experimental simulation with global fire emission database; AKFED; Alaskan Fire Emissions Database