# Peer review of "Forcing Global Fire Emissions Database burned-area dataset into Community Land Model version 5.0: Impacts on carbon and water fluxes at high latitudes"

_Geoscientific Model Development, 2022_

## Author Comment (AC1)

**Response Letter**

**[RC1: Referee #1]**

The authors were trying to evaluate the impacts of using prognostic and diagnostic wildfire schemes on ecosystem carbon and hydrological cycles simulated by CLM5-BGC. They found the default CLM5-BGC overestimated/underestimated the burned area in Eastern Siberia/Alaska, causing the overestimating/underestimation of wildfire carbon emissions. In contrast, the CLM5-BGC prescribed with observational burned area showed evident improvement in simulating wildfire carbon emissions for both regions. They further compared the two simulations with different wildfire schemes, in terms of major carbon and water fluxes, and showed larger influences on NEE than those for other variables. The modeling idea was unique, in terms of prescribing remote-sensing burned area within land surface model, especially across the high latitudes regions. However, the model evaluations and intercomparisons still need to be improved. For example, the authors may compare these two model results for more variables (e.g., LAI, GPP, ET, Soil Moisture) using different sources of observations or observation-based products. The authors could also add more thoughts/analyses on thow to quantify and reduce wildfire related biases for CLM5-BGC, in terms of drivers, processes and parameters.

Response: Thank you for your valuable inputs and insights into our article. In this study, we found that the application of the GFED burned area into CLM5-BGC caused significant changes in carbon emission and NEE. Therefore, we added the AKFED carbon emissions and GEOS-Carb CASA-GFED NEE to show the improvement. However, the other fluxes (GPP, NPP, and ET) and LAI did not change considerably after application of the GFED burned area. We discussed that, because the ratio of the fire area to the total area was relatively small, the LAI after fires was not substantially different due to the small fire area compared to the total area (LL). In other words, it is difficult to expect an improvement in LAI, GPP, NPP, and ET through the application of GFED. We, therefore, believe that a direct comparison to direct GPP, NPP, LAI, and ET products is not necessary and beyond the scope of our study.

As per this suggestion from the reviewer, we have added a detailed discussion about the limitations and future direction of the CLM fire module, as shown below:

LL 216: *In addition, wildfires are strongly affected by the weather conditions after the fire ignition. For example, wind and precipitation determine spread and duration of the fire. However, in CLM5-BGC, the fire ignition and fire spread rate are simultaneously calculated based on the weather conditions of fire ignition or pre-fire. Moreover, the persistence of each fire is assumed to be equal to 1 in CLM5-BGC. However, fires can last longer depending on climate conditions, which increase the burned area. Therefore, fire dynamics depending on the weather conditions after the fire ignition are necessary to reduce the biases in fire calculations.*

LL 285: *As the same fractional area burned is imposed on each PFT in a grid, the simulated carbon emission could be different from observed carbon emission. For example, when an*

*observation of forest fire is applied into CLM5-BGC, the fractional area burned is imposed on both grasses and trees in the same grid, causing biases in the carbon emission values. Therefore, a reasonable method of imposing grid-level burned areas into the PFT-level is required.*

Additionally, sentences between lines 231 and 234 read confusing;

Response: Thank you for pointing this issue out. We have revised them for clarity, as shown below:

LL 237: *Moreover, the correlations between the simulated carbon emissions and AKFED carbon emissions were determined (OL: 0.31 and EXP-GFED4: 0.96). While the root mean square error (RMSE) between the simulated carbon emissions and the AKFED carbon emissions decreased after applying the GFED4 burned area (OL: 20.48 g m$^{-2}$ year$^{-1}$ and EXP-GFED4: 10.98 g m$^{-2}$ year$^{-1}$), the RMSE between the simulated carbon emissions and the GFED4 carbon emissions increased (OL: 11.02 g m$^{-2}$ year$^{-1}$ and EXP-GFED4: 20.93 g m$^{-2}$ year$^{-1}$).*

and color scheme in Fig. 3 needs to be reversed?

Response: Thank you for your valuable suggestion regarding this figure. Based on your suggestion, we have revised the color in Fig. 3, as shown below:

[Figure]

**[RC2: Referee #2]**

Hocheol Seo and Yeonjoo Kim integrated GFED burned area dataset into CLM5-BGC model and investigated how fire activity affect ecosystem carbon and water fluxes over Alaska and Easter Siberia. They found that using GFED observed burned area, CLM5-BGC performed better in capturing fire emissions. Moreover, the carbon emissions over Alaska was sensitive to wildfire, while transpiration over Easter Siberia was insensitive to burn. The paper is well constructed. Below are major concerns:

The design of the model experiment needs to be improved to fully account the impact of historical fires and for a fair comparison between OL vs EXP-GFED4 runs.

This study used BGC version of CLM5, that have carbon, nutrient, water cycles. The 200 year spin up might be enough to stabilize soil temperature and moisture, but is too short to stabilize soil carbon pool and ensure a quasi-steady state condition. Suggestion: plot out total ecosystem carbon for the last 10 or 20 years of spinup period, the changes of total ecosystem carbon should be trivial. Also, long-term average of net ecosystem exchange (NEE) should be near zero in the end of spinup. If not, tried a longer spinup period.

After the spinup, a long-term transient simulation (starting from year 1850 or 1901) is necessary to ensure the land use, warming, and CO2 enrichment signals are all appropriately picked up by the CLM5-BGC model. For example, in this period, we often see initial decline of vegetation carbon due to land use, and then enhanced vegetation growth in response to warming and higher CO2 concentration will overwhelm. Such historical changes of vegetation activities, soil moisture conditions will affect fuel availability and combustibility for simulations from year 2001-2012 (period of focus for analysis). Otherwise, without appropriate spinup and transient simulation, the comparison of fire emissions and transpiration fluxes and others might not be convincing.

Response: Thank you for your valuable suggestion regarding our article. We did the spin up simulation for 200 years using the initial file, which already stabilized the carbon and water storage of the year 2000 to an equilibrium state. This initial file, which was provided by NCAR, makes us skip both long-term spin-up simulations and transient runs. As there may be stabilization issues caused by differences in resolution, spin-up simulations for an additional 200 years were performed. We have revised the manuscript to specify the spin-up process, as shown below:

LL 174: *Figure 2 shows the spin-up simulation, which stabilizes the land state, including the LAI, soil moisture, and soil temperature, with the initial file of the year 2000 in the equilibrium state. It was repeatedly run for 200 years using 20-year CRU/NCEP forcing data for 1980–2000 before adopting OL and EXP-GFED4.*

Furthermore, we verified that the carbon and soil moisture was enough for stabilization, and the figure below shows that the NEE was near zero during the spin-up period. In addition, we show the value of Fuel C mapped over high latitudes. In most cases of high latitudes, Fuel C exceeded 1050 $gC/m^2$, which is the upper fuel threshold. This means that there was enough of

Fuel C to burn during the CLM simulation period. Please refer to Equation 2.24.6 in Lawrence et al. (2019).

[Figure]

● CLM5-BGC baseline model performance and improvement

In order to understand how much of improvement was due to the integration of GFED burned area dynamics, it will be necessary to first showed CLM5-BGC baseline model performance over the area of interest. It will be good to compare OL and EXP-GFED4 simulations against observations (Figure 6-10). Suggested datasets are e.g., FLUXCOM GPP/NEP, MODIS LAI, GLEAM ET, GEOCARBON vegetation biomass.

Response: Thank you for your valuable inputs and insights into our article. In this study, we found that the application of the GFED burned area into CLM5-BGC caused significant changes in carbon emission and NEE. Therefore, we added the AKFED carbon emissions and GEOS-Carb CASA-GFED NEE to show the improvement. However, the other fluxes (GPP, NPP, and ET) and LAI did not change considerably after application of the GFED burned area. We discussed that, because the ratio of the fire area to the total area was relatively small, the LAI after fires was not substantially different due to the small fire area compared to the total area (LL 264). In other words, it is difficult to expect an improvement in LAI, GPP, NPP, and

ET through the application of GFED. We, therefore, believe that a direct comparison to direct GPP, NPP, LAI, and ET products is not necessary and beyond the scope of our study.

● Area of focus

The box area in Figure 1 seems arbitrary. It will be better to use geographic boundary (e.g., state of Alaska) instead of a random box.

Response: Thank you for this valuable advice. We have revised the mistake in the domain description, as shown below. We chose the boundary of "Interior Alaska" (200° E–218° E and 61° N–70° N), where fires occur frequently. However, the burned area is underestimated over Interior Alaska. Therefore, the effect of an accurate fire can be clearly shown through a comparison of the two simulations (OL and EXP-GFED4) in this boundary. Boundary b [Eastern Siberia (130° E–148° E and 61° N–70° N)] was chosen because the fire simulation was too overestimated even though both domains have the same size and latitudes.

LL 160: *In this study, we focused on Alaska (200° E–218° E, 61° N–70° N) and Eastern Siberia (130° E–148° E, 61° N–70° N), which are located at northern high latitudes (Figure 1). Both domains have the same size and latitudes.*

LL 495: *Figure 1. Study domain, (a) Alaska (200° E–218° E and 61° N–70° N), and (b) Eastern Siberia (130° E–149° E and 61° N–70° N).*

● Plant functional type differences

It is not clear, how each different plant functional type (PFT) handled when GFED data is integrated to CLM5-BGC at gridcell level. At each gridcell, CLM has multiple PFT that have different level of fuel conditions (e.g., arctic grass vs boreal tree). When the observed burned area gets integrated, how to reasonably assign the burn to each PFT? For example, the observed burn may occur over forest, while in CLM the observed burn is imposed on the whole gridcell that have both trees and grasses.

Response: Thank you pointing this out. The burned area was calculated at the grid level, not at the PFT-level. Once a grid-level burned area is calculated, the same fractional area burned (/s) is imposed on each PFTs in the grid. We have added this explanation in Section 2.1 and a related discussion about this, as shown below:

LL 92: *In CLM, the burned area is calculated at the grid level and the fire emissions are calculated at a PFT level. Once a grid-level burned area is calculated, the same fractional area burned is imposed on each PFT in the grid. The PFT-level carbon emission from the fire is calculated as follows (24.26 in Lawrence et al., 2019):*

$$CE = A \cdot C \cdot CC \qquad\qquad (1)$$

*where CE is the carbon emission; A is the fractional area burned; C is a vector with the carbon density of leaves, stems, and roots, carbon transfer, and carbon pools; and CC is the corresponding combustion completeness factor vector.*

LL 285: *As the same fractional area burned is imposed on each PFT in a grid, the simulated carbon emission could be different from observed carbon emissions. For example, when an observation of forest fire is applied into CLM5-BGC, the fractional area burned is imposed on both grasses and trees in the same grid, causing biases in the carbon emission values. Therefore, a reasonable method of imposing grid-level burned areas into the PFT-level is required.*

Temporal scale issue
GFED is a monthly product, while CLM5-BGC runs at a much finer temporal resolution (e.g., 30min). How to assigned monthly burned area to each individual time step in CLM5-BGC? It will make a big difference to apply the burned area to the beginning versus to the end of each month?

Response: The GFED4 product is provided at a $0.25° \times 0.25°$ resolution and at daily and monthly temporal resolutions. Thus, we applied the daily scaled GFED products to CLM. This daily data is equally divided into a half-hourly model timestep. Please refer to LL 181 in the manuscript.

**[CC1: Sarah Gallup]**

This study focuses on a decidedly useful topic. The design is a reasonable approach to learning about the limitations and potential to improve CLM's fire simulations. Phrasing and copyediting are substandard. The analysis of the results needs more meat in terms of using details of the two runs to better understand how and why they differ. The model runs would support a substantially tighter and more coherent assessment. What can the authors show or even speculate about why CLM fire matches not only rather poorly to the datasets, but also differently in the two continents? Ssaying CLM Fire is "limited", pointing out that it is imperfect, is less useful than helping the community think about reasons and specifics. Several of the speculations about real-world reasons the CLM fire algorithm is imperfect are insightful and useful.

Response: Thank you for raising this concern. As per the reviewer's suggestion, we have added a detailed discussion about the limitations and future direction of the CLM fire module, as shown below:

LL 216: *In addition, wildfires are strongly affected by the weather conditions after the fire ignition. For example, wind and precipitation determine spread and duration of the fire. However, in CLM5-BGC, the fire ignition and fire spread rate are simultaneously calculated based on the weather conditions of fire ignition or pre-fire. Moreover, the persistence of each fire is assumed to be equal to 1 in CLM5-BGC. However, fires can last longer depending on climate conditions, which increase the burned area. Therefore, fire dynamics depending on the weather conditions after the fire ignition are necessary to reduce the biases in fire calculations.*

LL 285: *As the same fractional area burned is imposed on each PFT in a grid, the simulated carbon emission could be different from observed carbon emissions. For example, when an observation of forest fire is applied into CLM5-BGC, the fractional area burned is imposed on both grasses and trees in the same grid, causing biases in the carbon emission values. Therefore, a reasonable method of imposing grid-level burned areas into the PFT-level is required.*

Some notes about uncertainty in the benchmark datasets seem warranted. As an obvious example, GFED emissions too are a model. While it is reasonable to assume the comparison data is more accurate than an ESM simulation of fire, what considerations about the inevitably imperfect inventories should a reader keep in mind? How similar are the two inventories' derivation algorithms and data sources? Making the comments specifically relevant to the patterns the study finds would be most helpful. As only an example, what is the correlation of GFED and AKFED emissions for the study area and period?

Response: Thank you for providing this valuable suggestion. Indeed, GFED and AKFED emissions are also model outputs. The correlation between GFED and AKFED emissions over Alaska is 0.9. However, when the fire module of CLM5-BGC was developed, the global-scale GFED carbon emissions were benchmarked (Li et al. 2012). Therefore, we compared the

simulated carbon emissions and the GFED emissions. We also performed an additional analysis with AKFED emissions to reduce the uncertainties.

Any information about the relevance of peat fire would make this paper a substantially stronger tool for improving fire in CLM. As examples, what is the relative abundance of peat in the study area compared to the rest of Siberia? What portion if any of the "open loop" Siberian burned area and emissions were generated from the peat fire algorithm within CLM fire?

Response: Thank you for this valuable suggestion. Peat fires are becoming increasingly important. However, simulating peat fires using CLM is insufficient. We verified that the peatland proportion of our domain in CLM was too low (Alaska: 0%, Eastern Siberia: 2%); hence, we could ignore the effects of peatland fires. We believe that the distribution of peatlands would be detected more accurately and employed primarily in the model to study the impacts of peat fires.

Thank you for tackling this study.
general - Please either use a consistent number of significant digits, or justify why not.
Response: Thank you for raising this concern and the valuable suggestion. As per the reviewer's suggestion, we have done the following revisions in the manuscript:

LL 256: *In Alaska (Fig. 6a), the difference in LAI between OL and EXP-GFED4 was the largest in 2005 (0.03 m²/m²).*
LL 314: *The average change rates of NEE and carbon fluxes were 4914 gC ha⁻¹ and 4881 gC ha⁻¹ and 771 gC ha⁻¹ and 797.6 gC ha⁻¹ in Alaska and Eastern Siberia, respectively.*

line 41 - 'Human-caused' is conflated with human-ignited. Warmer climate, too, is human-caused.

Response: We truly appreciate your valuable insight in this matter. As per the reviewer's suggestion, we have revised this in the manuscript, as shown below:

LL 41: *Fires at high latitudes are primarily ignited by natural processes rather than by humans.*

102 - Equations 1 & 2 should be cited, including with equation numbers from Lawrence19

Response: Thank you for pointing this out. As per the reviewer's suggestion, we have added following lines in the manuscript:

LL 93: *The PFT-level carbon emission from the fire is calculated as follows (24.26 in Lawrence et al., 2019)*
LL 103: *In CLM5-BGC, the amount of leaf carbon to litter (Ψ) caused by fire is calculated as follows (24.27 in Lawrence et al., 2019)*

106 - By "leaf size" do you mean LAI? The terms are not interchangeable.

Response: Thank you for raising this valid concern. As per the reviewer's suggestion, we have done the following revision in the manuscript:

LL 109: *The leaf area index (LAI) is recalculated based on the adjusted amount of leaf carbon. In addition, the methods by which the amount of carbon in live stems, dead stems, and roots and the storage pool are adjusted due to fires are similar to those mentioned above.*

153 - What data source do you use for woodfuel burning estimates?

Response: Thank you for pointing this issue out. As per the reviewer's suggestion, we have added a description in the manuscript.

LL 157: *FuelE is the carbon emissions from wood-fuel burning in GEOS-Carb CASA-GFED.*

166 - Pls explain why you chose the specific area within Siberia, and what relevant ways it is similar to or different from the rest of Siberia.

Response: We appreciate you asking about this. Boundary b (Eastern Siberia (130°E–148° E and 61° N–70° N)) was chosen because the fire simulation was too overestimated even though both domains have the same size and latitudes.

201 - an egregious example of the need for copyediting. Ditto l. 230-234.

Response: Thank you for pointing this out. As per the reviewer's suggestion, we have revised this in the manuscript, as shown below:

LL 237: *Moreover, the correlations between the simulated carbon emissions and AKFED carbon emissions were determined (OL: 0.31 and EXP-GFED4: 0.96). While the root mean square error (RMSE) between the simulated carbon emissions and the AKFED carbon emissions decreased after applying the GFED4 burned area (OL: 20.48 g m$^{-2}$ year$^{-1}$ and EXP-GFED4: 10.98 g m$^{-2}$ year$^{-1}$), the RMSE between the simulated carbon emissions and the GFED4 carbon emissions increased (OL: 11.02 g m$^{-2}$ year$^{-1}$ and EXP-GFED4: 20.93 g m$^{-2}$ year$^{-1}$).*

217 - While the general point is very well taken, "no human impacts" is an overstatement. See p.29 of https://fire.ak.blm.gov/content/aicc/Alaska%20Statewide%20Master%20Agreement/3.%20Alaska%20Interagency%20Wildland%20Fire%20Managment%20Plan%20(AIWFMP)/2022%20AIWFMP%20Final%20Signed%202022-02-28.pdf. Responding agencies will "conduct site protection as warranted."

Response: We thank you for raising this valid concern. As suggested, we have removed this sentence from the manuscript.

237 - OK, but there now exists information about differences between GFED3 and GFED4. To what extent is Veraverbeke's explanation that you reiterate perhaps now addressed – or not?

Response: Thank you for raising this concern. We showed that GFED products have a weakness in representing the carbon emissions over Alaska. We referred to the Veraverbeke's explanation, which showed that there was a possibility that GFED3s underestimated carbon combustions which can be attributed to the presence of fires in boreal forest regions, and we thought this was similar in GFED4.

239 - Rather than speculate, pls look up the numbers and compare them at least to each other and ideally also to additional references.

Response: Thank you for this valuable suggestion. As per the reviewer suggestion, we added the values of the combustion completeness factors in CLM5-BGC, as shown below. We have also added a reference about the combustion completeness factor values in boreal forests.

LL 246: *The combustion completeness factor for leaves is 0.8 and that for stems ranges from 0.27 to 0.8, depending on the PFTs in the CLM5-BGC. According to van der Werf et al. (2010), the combustion completeness of aboveground live biomass, which ranges 0.3~0.4 in the boreal region, is lower than that in other regions. Therefore, the combustion completeness factors for boreal trees may be lower than the current default value in CLM5-BGC.*

Reference: *van der Werf, G. R., Randerson, J. T., Giglio, L., Collatz, G. J., Mu, M., Kasibhatla, P. S., Morton, D. C., Defries, R. S., Jin, Y., and van Leeuwen, T. T.: Global fire emissions and the contribution of deforestation, savanna, forest, agricultural, and peat fires (1997-2009), Atmospheric Chemistry and Physics, 10(23), 11707–11735, https://doi.org/10.5194/acp-10-11707-2010, 2010.*

256 - rates of change, or changes?

Response: Thank you for pointing this error out. We have corrected it to "rates of changes."

279 - 281 needs replacing. Line 279 is an overstatement; Line 280 was known before the study started simply because all models are imperfect; Line 281 is not a logical conclusion based on the prior two statements. Writing a stronger analysis as requested in the general notes above will provide better material to summarize in this paragraph.

Response: Thank you for your valuable insights and suggestions on this section. As per the reviewer's suggestion, we have revised this sentence, as shown below:

LL 290: *As CLM5-BGC is still limited in representing fire processes, there is a large difference in the burned area between the simulation and observation. By comparing our experiments of OL and EXP-GFED4 in Alaska and Eastern Siberia, we identified the effects of accurate fire simulation on carbon fluxes over Alaska and Eastern Siberia.*

---

## Editor Decision (ED1)

Major comments:

- Did your spinup include land use areas? (How much land use even is there in the study regions?)
- CC1 response a bit lacking, as they raised a good point about peat fires. Even if you determine it's not an issue, you should mention this in the MS.
- Throughout: The paper is framed as forcing CLM with GFED4 burned areas to learn about real-world hydrological and biogeochemical cycling. However, the analyses make it more of a "model evaluation" paper—you're mostly seeing how much CLM's results are affected by its biased burned area, rather than learning much about real-world fire. You should either reconsider the framing (preferable to me —it's easier, and still an important evaluation!) or add analyses supporting your original framing.
- Results, throughout: Contextualize numbers. I don't have an intuitive sense of what, e.g., a difference of 5.41 mm in canopy evaporation means. What do these results mean in terms of percent difference, either between simulations/observations or between simulations? Are they biogeochemically/ecologically meaningful differences? (You don't need to give % change for every number, although in some cases that might help. What I'm saying is, you need to give the reader a better sense of what the numbers *mean* in relative terms.)
- It's good that you discuss possible reasons for the burned area and C flux results. However, these discussions are important enough (and will be long enough, once you expand them as I suggest below) that they should be moved out of the Results and into a new Discussion section.

Detailed comments:

- Throughout: Why is the default run called "open-loop"? Wouldn't it be more straightforward to just call it "default"?
- Title:
  - "Global Fire Emissions Database burned-area dataset into Community Land Model version 5.0" doesn't work. Maybe something like, "Forcing the Community Land Model version 5.0 with burned area from the Global Fire Emissions Database"?
  - You can delete " – Biogeochemistry" to make the title simpler.
- Sections 2 and 3 should be combined into a single "Methods" section. It especially doesn't make sense to have Sect. 2.2 in a different section from the rest of the experimental design. Also, Sect. 3 has the same name as Sect. 3.2. Here's my suggested reordering under a single "Sect. 2, Methods", new ← original:
  - 2.1 ← 2.1: Model description
  - 2.2 ← 3.1: Site description
  - 2.3 ← 3.2: Experimental design
  - 2.4 ← 2.2: Fire and C fluxes datasets
- L10: Capitalize "Global Fire Emissions Database" and "Community Land Model"
- L14: "trends" should be "signs" or "directions"
- L26: Delete "remarkably"—too opinionated
- L31: Replace ", at" with "in"

- L38: "regions" is weird here, since it usually refers to geographical areas. Try replacing "carbon in belowground regions" with "belowground carbon".
- L71: Capitalize "Community Earth System Model"
- L75: Hyphen needed in "sub grid"
- L84: Capitalize "Lightning Imaging Sensor" and "Optical Transient Detector"
- L94, 104: What do "24.26" and "24.27" refer to? Equation numbers? I don't see those anywhere in Lawrence et al. (2019).
- L96 (Eq. 1), L106 (Eq. 2): All vectors ($CE$, $C$, $CC$, $f_i$, $M$) should be in boldface italics: [https://www.geoscientific-model-development.net/submission.html#math](https://www.geoscientific-model-development.net/submission.html#math)
- L114:
    - Delete "Especially, the "
    - "represent" should be "represents"
- L127: "leaf size" should be "leaf area"
- L141–145
    - Is Veraverbeke et al. (2015) the citation for AKFED? If so, cite that in the first sentence here. If not, add the correct citation—and then why is Veraverbeke et al. (2015) discussed at all?
    - L144: "presumed" is almost certainly not the right word. "Calculated"? "Determined"?
- L146: Not just EXP-GFED4, right? Also OL?
- L169: Replace "but" with "and"
- L195: Replace "big fires" with ""large burned areas or "anomalous years"; "big fires" implies individual contiguous burn patches, which may not be the case.
- L199: What is this sentence trying to say? Why "especially"? What's special about it?
- Paragraph at L201–5 needs a total rework.
    - "inadequately" is a value judgment; whether the model performs adequately depends on what question it's being used to answer. Replace this with something that describes the CLM bias objectively.
    - L201: Re-state grid cell resolution here.
    - L202: Observed by GFED? Or AKFED?
    - L202–3:"a few"? How many?
    - L204: "simulating largely burned areas"? What does this mean?
    - L204: "more grid cells"? Relative to what?
- Paragraphs at L207–26 need rework.
    - Please combine these paragraphs. You do some discussion in the second paragraph, which is confusing because the paragraph break makes it seem like you're moving on to something else.
    - You should also discuss the issues with wind speed in global fire models: Lasslop et al. (2015), [https://www.publish.csiro.au/wf/WF15052](https://www.publish.csiro.au/wf/WF15052)
    - "position"?

- "misunderstanding" is not the right word. Do you mean "misrepresentation"?
- What do you mean by "the limitation of using point data in the grid-based model"?
- L219: "Persistence" ("duration" would be clearer) has what units?
- L220: "fires can last longer" in CLM or real life?
- Expand discussion of fire duration into its own paragraph and add citations. Much literature exists about both (a) real-world fire durations (especially in Alaska, where large fires contribute a huge proportion of burned area), (b) the effect of the constant-duration (or max 1 day) assumption in fire models, and (c) the effect of including dynamic, > 1 day fire duration in models.
- Paragraph about Alaskan fire policy needs expansion. What do those different levels mean? How much area is in each level, especially in your study area?
- L224: "anthropophonic" is not a word. "anthropogenic"?

- It's unclear what the difference is between Sections 4.2 and 4.3. You should strongly consider combining them to tell a more cohesive story about your results.

- L241–9
  - Did C emissions change much between what Veraverbeke et al. looked at (GFED3s) and GFED4?
  - Be clearer throughout about when you're discussing CLM vs. GFED (vs. real life?) combustion completeness factors.
  - You cite the combustion completeness factors for GFED3 (van der Werf et al., 2010) instead of the dataset you actually used (GFED4; Giglio et al., 2013). There were actually important changes to how combustion completeness works in GFED4!
  - L248: Tilde should be an en dash

- L251: "form" should be "from"

- L253:
  - "more reliable" in what? GFED/CASA or CLM?
  - "dominant" in what? Observations and/or GFED/CASA and/or CLM?

- L256–61: This paragraph feels weird in a section about carbon fluxes without you first having discussed GPP/NEP/NPP. LAI is an explanatory factor of those things and thus should go after the GPP/NEE/NEP discussion.

- L264: "rate" is not correct here, as it implies something with time in the denominator. Replace "rates of changes" with "differences".

- L290–6: This paragraph fits more in the Conclusions section.

- L321–31:
  - At some point this paragraph transitions from talking about both regions to just Alaska. Make Alaska its own paragraph, as you did for Siberia.
  - L322: Replace "grids" with "grid cells".
  - L323: "affected" in what direction?
  - L324: "may"?
  - L329: Replace "the CLM-dynamic global vegetation model" with just "CLM"

- L370: Please cite the specific version (git tag or commit SHA) of CLM on which you made your changes.

- L465–6: Please replace citation with Rabin et al. (2018): https://gmd.copernicus.org/articles/11/815/2018/

- Figs. 2, 3, 5, 6, 9, 10: Please use more colorblind-friendly colors in Fig. 5, especially avoiding red and green. For all these figures, using different line styles (solid vs. dashed vs. dotted) and/or a variety of markers (instead of just squares) would also help. Some useful resources can be found here: https://www.geoscientific-model-development.net/submission.html#figurestables

---

## Author Response (AR2)

**Response Letter**

**Editor:**

Major comments:

Did your spinup include land use areas? (How much land use even is there in the study regions?)

>> CLM includes multiple land units, including lakes, urban, glaciers, vegetation, and crops. Although we considered all land units in our simulations, the natural vegetation land unit was found to occupy the largest area, and the lake, urban, and glacier land units occupied less than 1% in both regions. Therefore, the influences of the lake, urban, and glacier units were extremely small in our simulations. We have added this in the revised manuscript.

L140: *Notably, the largest areas were the natural vegetation and crop land units, and the lake, urban, and glacier land units occupied less than 1% in both regions.*

CC1 response a bit lacking, as they raised a good point about peat fires. Even if you determine it's not an issue, you should mention this in the MS.

>> First, the impact of peat fire in our simulation was quite minimal because the peatland fractions in CLM, which were derived from three datasets (Olson et al., 2001; Tarnocai et al., 2011; Lehner and Döll, 2004), were low over both regions. (Alaska: 0%, Eastern Siberia: 2%). Recently, however, peat fires and even smouldering wildfires have been occurring frequently, becoming an issue in Alaska and Eastern Siberia (Scholten et al., 2021). We, therefore, discussed that peat fires should be considered to improve the coverage of peatland in the future, in the revised manuscript.

L335: *Moreover, inaccurate coverage of peatland can also cause a bias in burned area calculations. Peat fire and along with smouldering fire have been reported over both regions for several years (Scholten et al., 2021). However, peat fire was barely simulated in CLM-BGC5 because the fractions of peatland, which were derived from three datasets (Olson et al., 2001; Tarnocai et al., 2011; Lehner and Döll, 2004), were low over both regions. (Alaska: 0%, Eastern Siberia: 2%). On the contrary, several studies reported that there is sufficient coverage of peatland in both areas to consider the existence of peatland fires (Yu et al., 2010; Qiu et al.,*

*2019). For instance, the coverage of peatland is 72–168 $10^3 km^2$, and 16–32 Pg of carbon is stocked in peatland in Alaska. Therefore, to simulate peat fires accurately, an improvement of the dataset used for peatland coverage in CLM should be considered.*

Olson, D. M., Dinerstein, E., Wikramanayake, E. D., Burgess, N. D., Powell, G. V. N., Underwood, E. C., D'Amico, J. A., Itoua, I., Strand, H. E., Morrison, J. C., Loucks, C. J., Allnutt, T. F., Ricketts, T. H., Kura, Y., Lamoreux, J. F., Wettengel, W. W., Hedao, P., and Kassem, K. R.: Terrestrial Ecoregions of the World: A New Map of Life on Earth, BioScience, 51, 933–938, 2001.

Tarnocai, C., Kettles, I. M., and Lacelle, B.: *Peatlands of Canada Map.* Geological Survey of Canada, Open File 3834. Scale 1: 6 500 000, Natural Resources Canada, Ottawa, 2000.

Lehner, B. and Döll, P.: Development and validation of a global database of lakes, reservoirs and wetlands, J. Hydrol., 296, 1–22, https://doi.org/10.1016/j.jhydrol.2004.03.028, 2004.

Qiu, C., Zhu, D., Ciais, P., Guenet, B., Peng, S., Krinner, G., Tootchi, A., Ducharne, A., and Hastie, A.: Modelling northern peatland area and carbon dynamics since the Holocene with the ORCHIDEE-PEAT land surface model (SVN r5488), Geosci. Model Dev., 12, 2961–2982, https://doi.org/10.5194/gmd-12-2961-2019, 2019.

Scholten, R. C., Jandt, R., Miller, E. A., Rogers, B. M., and Veraverbeke, S.: Overwintering fires in boreal forests, Nature, 593, 399–404, 2021.

Throughout: The paper is framed as forcing CLM with GFED4 burned areas to learn about real-world hydrological and biogeochemical cycling. However, the analyses make it more of a "model evaluation" paper—you're mostly seeing how much CLM's results are affected by its biased burned area, rather than learning much about real-world fire. You should either reconsider the framing (preferable to me —it's easier, and still an important evaluation!) or add analyses supporting your original framing.

>> As per the editor's suggestion, we have re-organized the revised manuscript to include a separate section discussing the discrepancy between the model and real-world fire.

1) Ignition process of fires and persistence

L305: *First, the limited representation of fire ignition sources and spread may create*

*discrepancies between modeled and observed burned areas. Lightning, which is a major source of fire at high latitudes, especially in Alaska, has increased because of the warming climate (Kępski and Kubicki, 2022). Although the lightning frequency at high latitudes varied yearly, the climatology of the 3-hourly lightning frequency from 1995 to 2011 was used in CLM. Moreover, the calculated ratio of cloud-to-ground lightning has large uncertainties and may cause models to misestimate fire ignition and burned areas. Furthermore, it is inherent that the grid-based large-scale model is limited in capturing micro-environmental impacts on fire spread. Fires spread differs depending not only on the temperature, precipitation, wind speed, and direction but also on the composition of vegetation at the local scale.*

2) Fire duration

L313: *In addition, wildfires are strongly affected by the weather conditions after the fire ignition. For example, wind and precipitation determine the spread and duration of fire. However, in CLM5-BGC, the fire ignition and fire spread rate are simultaneously calculated based on the weather conditions of fire ignition or pre-fire. Moreover, wildfires in ecosystems persist from hours to months, depending on ecosystem characteristics and climate conditions. However, the duration of each fire is assumed to be equal to one day in CLM5-BGC (Li et al., 2012). For example, Andela et al. (2019) reported that the average fire duration in a boreal forest was longer than those in other regions, and the average size of each fire in the boreal forest was larger than those in temporal forests and under deforestation. Moreover, wind speed is an important factor determining fire spread in the model. In CLM, the spread of fire increases as the wind speed increases. However, according to Lasslop et al. (2015), there is strong variation in the burned fraction with wind speed, characterized by an increase until a certain wind speed threshold is reached and a decrease thereafter. The study suggests that global fire models should avoid a strong amplification for higher wind speeds to prevent overestimation of modelled burned areas.*

3) Fire policy management

L326: *The management system and infrastructures for fires vary by country or region. For instance, there are four types of fire policy options in Alaska, namely critical, full, modified, and limited, according to the levels of anthropogenic effort in extinguishing the fire (Phillips et al., 2022). For example, fire suppression is the highest priority at the critical protection level*

*because wildfire can threaten human life and inhabited property. The lowest priority for fire-related resource assignments is applied at the limited protection level. In Alaska, areas under the full, modified, and limited management options occupy 16%, 16%, and 67% of Alaska, respectively. Critical-protection-level areas occupy less than 1% of Alaska. In CLM5-BGC, however, the suppression impact is calculated based on the GDP and population, which may underestimate burned areas in the limited regions of Alaska because of the large GDP of the United States.*

Results, throughout: Contextualize numbers. I don't have an intuitive sense of what, e.g., a difference of 5.41 mm in canopy evaporation means. What do these results mean in terms of percent difference, either between simulations/observations or between simulations? Are they biogeochemically/ecologically meaningful differences? (You don't need to give % change for every number, although in some cases that might help. What I'm saying is, you need to give the reader a better sense of what the numbers mean in relative terms.)

>> As per the editor's suggestion, we have added the percent differences and their implications in the revised manuscript.

L274: *We observed that more rainfall reaches the ground, which would make the ground evaporation rate higher in regions with more burned areas, especially in 2004 and 2005 in Alaska. The differences in annual canopy evaporation, canopy transpiration, and ground evaporation between the two simulations were 5.41 mm and 13.37 mm, 2.3 mm and 6.26 mm, and −1.39 mm and −7.4 mm in 2004 and 2005, respectively. Canopy transpiration decreased by 3%, canopy evaporation decreased by 12%, and ground evaporation increased by 10% in 2004 and 2005 after applying the GFED4 burned area into CLM. This is consistent with the findings of Li et al. (2017) and Seo and Kim (2019), showing that canopy evaporation and canopy transpiration decreased and ground evaporation increased when comparing the simulation with and without fire. Furthermore, the total ET in the presence of fire decreased by 6.32 mm and 12.08 mm in 2004 and 2005, respectively, indicating that canopy evaporation is more strongly influenced by fires over Alaska in CLM.*

*In Eastern Siberia, the patterns of canopy evaporation and ground evaporation were the same as those of Alaska. Canopy evaporation increased and ground evaporation decreased in EXP-GFED4 because the simulated burned area decreased, which was noticeable from 2009 to 2012*

*(Fig. 9f and 9h). However, the canopy transpiration of EXP-GFED4 was similar to that of CLM-Default. In other words, there was no significant change in canopy transpiration due to a change in burned area. Furthermore, the ET with the burned area applied changed slightly in Eastern Siberia. Differences in the average canopy evaporation and ground evaporation were −9.19 mm (28%) and 6.97 mm (10%) from 2009 to 2012, respectively. The reasons for the smaller change in canopy transpiration is related to soil moisture and leaf size.*

It's good that you discuss possible reasons for the burned area and C flux results. However, these discussions are important enough (and will be long enough, once you expand them as I suggest below) that they should be moved out of the Results and into a new Discussion section.

\>\> As per the editor's suggestion, we have added a discussion section in the revised manuscript. It includes several discussions related to fire duration, wind speed, peat fires, carbon fluxes, and others.

1) Fire duration

L313: *In addition, wildfires are strongly affected by the weather conditions after the fire ignition. For example, wind and precipitation determine the spread and duration of fire. However, in CLM5-BGC, the fire ignition and fire spread rate are simultaneously calculated based on the weather conditions of fire ignition or pre-fire. Moreover, wildfires in ecosystems persist from hours to months, depending on ecosystem characteristics and climate conditions. However, the duration of each fire is assumed to be equal to one day in CLM5-BGC (Li et al., 2012). For example, Andela et al. (2019) reported that the average fire duration in a boreal forest was longer than those in other regions, and the average size of each fire in the boreal forest was larger than those in temporal forests and under deforestation. Moreover, wind speed is an important factor determining fire spread in the model. In CLM, the spread of fire increases as the wind speed increases. However, according to Lasslop et al. (2015), there is strong variation in the burned fraction with wind speed, characterized by an increase until a certain wind speed threshold is reached and a decrease thereafter. The study suggests that global fire models should avoid a strong amplification for higher wind speeds to prevent overestimation of modelled burned areas.*

2) Wind speed

L320: *Moreover, wind speed is an important factor determining fire spread in the model. In CLM, the spread of fire increases as the wind speed increases. However, according to Lasslop et al. (2015), there is strong variation in the burned fraction with wind speed, characterized by an increase until a certain wind speed threshold is reached and a decrease thereafter. The study suggests that global fire models should avoid a strong amplification for higher wind speeds to prevent overestimation of modelled burned areas.*

3) Peat fires

L335: *Moreover, inaccurate coverage of peatland can also cause a bias in burned area calculations. Peat fire and along with smouldering fire have been reported over both regions for several years (Scholten et al., 2021). However, peat fire was barely simulated in CLM-BGC5 because the fractions of peatland, which were derived from three datasets (Olson et al., 2001; Tarnocai et al., 2011; Lehner and Döll, 2004), were low over both regions. (Alaska: 0%, Eastern Siberia: 2%). On the contrary, several studies reported that there is sufficient coverage of peatland in both areas to consider the existence of peatland fires (Yu et al., 2010; Qiu et al., 2019). For instance, the coverage of peatland is 72–168 103km2, and 16–32 Pg of carbon is stocked in peatland in Alaska. Therefore, to simulate peat fires accurately, an improvement of the dataset used for peatland coverage in CLM should be considered.*

4) Carbon fluxes – PFT distribution

L350: *The average change rates of NEE and carbon fluxes were 4914 gC ha$^{-1}$ and 4881 gC ha$^{-1}$ and 771 gC ha$^{-1}$ and 798 gC ha$^{-1}$ in Alaska and Eastern Siberia, respectively. The response of carbon emissions to fires was much more sensitive than those of GPP, NPP, and NEP; therefore, changes in carbon emissions are a major cause of the change in the NEE, which is consistent with previous results. Carbon release owing to wildfires was more sensitive in Alaska than Eastern Siberia under CLM5-BGC, as boreal trees are more distributed in Alaska than in Eastern Siberia. Based on the above results, we suggest that more accurate fire predictions are needed to understand ecosystem carbon fluxes, especially in Alaska.*

*Therefore, one can tell that the carbon fluxes were more sensitive in Alaska than in Eastern Siberia. The reasons for carbon emissions being more pronounced in Alaska than in Eastern Siberia could be explained by the vegetation distribution. The average ratio of carbon emissions to burned areas was 49.98 Tg Mha$^{-1}$ in Alaska and 9.76 Tg Mha$^{-1}$ in Eastern Siberia.*

*There was 95 Tg of leaf carbon and 8.3 Tg of live-stem carbon in Alaska and 29 Tg of leaf carbon and 2.4 Tg of live-stem carbon in Eastern Siberia in averages of CLM-Default and EXP-GFED4. Trees have a larger LAI and stems and thus more fuel combustibility and availability. Therefore, the ratio of carbon emissions to burned areas was high in forests than in grassland. Moreover, the final carbon fluxes between the atmosphere and vegetation were closely linked not only with vegetation metabolism but also with burned area and plant type. As the same fractional area burned is imposed on each PFT in a grid, the simulated carbon emission could differ from observed carbon emissions. For example, when an observation of forest fire is applied to CLM5-BGC, the fractional area burned is imposed on both grasses and trees in the same grid, causing biases in the carbon emission values. Therefore, a reasonable method of imposing grid-level burned areas into the PFT level is required.*

Detailed comments:

Throughout: Why is the default run called "open-loop"? Wouldn't it be more straightforward to just call it "default"? Title: "Global Fire Emissions Database burned-area dataset into Community Land Model version 5.0" doesn't work. Maybe something like, "Forcing the Community Land Model version 5.0 with burned area from the Global Fire Emissions Database"? You can delete " – Biogeochemistry" to make the title simpler.

>> As per the editor's suggestion, we have changed "open-loop" to "CLM-Default" and changed the title.

L1: *Forcing Global Fire Emissions Database burned-area dataset into Community Land Model version 5.0: Impacts on carbon and water fluxes at high latitudes*

Sections 2 and 3 should be combined into a single "Methods" section. It especially doesn't make sense to have Sect. 2.2 in a different section from the rest of the experimental design. Also, Sect. 3 has the same name as Sect. 3.2.

Here's my suggested reordering under a single "Sect. 2, Methods", new ← original: 2.1 ← 2.1: Model description /2.2 ← 3.1: Site description /2.3 ← 3.2: Experimental design 2.4 ← 2.2: Fire and C fluxes datasets

>> As per the editor's suggestion, we have changed the section number.

L10: Capitalize "Global Fire Emissions Database" and "Community Land Model"

>> We have changed these in the revised manuscript.

L9: *In this study, we employed the daily burned areas from satellite-based Global Fire Emission Database (version 4) (GFED4) into Community Land Model (version 5.0), with a biogeochemistry module (CLM5-BGC) to identify the effects of accurate fire simulation on carbon and water fluxes over Alaska and Eastern Siberia.*

L14: "trends" should be "signs" or "directions"

>> As per the editor's suggestion, we have changed them in the revised manuscript, as shown below.

L14: *The results showed that the simulated carbon emissions with burned areas from GFED4 (i.e., experimental run) were significantly improved in comparison to the open-loop run (that is default run), which resulted in opposite signs of the net ecosystem exchange for 2004, 2005, and 2009 over Alaska between the default and experimental runs.*

L26: Delete "remarkably"—too opinionated

>> We have deleted this in the revised manuscript.

L31: Replace ", at" with "in"

>>We have changed it in the revised manuscript.

L38: "regions" is weird here, since it usually refers to geographical areas. Try replacing "carbon in belowground regions" with "belowground carbon".

>> We have changed it in the revised manuscript, as shown below.

L38: *This could result in the release of belowground carbon, which can increase the levels of carbon dioxide in the atmosphere.*

L71: Capitalize "Community Earth System Model"

>> We have capitalized it in the revised manuscript.

L75: Hyphen needed in "sub grid"

>> We have revised it.

L84: Capitalize "Lightning Imaging Sensor" and "Optical Transient Detector"

>> We have capitalized them in the revised manuscript.

L94, 104: What do "24.26" and "24.27" refer to? Equation numbers? I don't see those anywhere in Lawrence et al. (2019).

>> As captured below, the equation is found in the Technical Description of version 5.0 of Community Land Model (CLM).

**24.1.3 Fire impact**

In post-fire regions, we calculate PFT-level fire carbon emissions from biomass burning of the $j$th PFT, $\phi_j$ (g C s$^{-1}$), as

$$\phi_j = A_{b,j}\mathbf{C}_j \bullet \mathbf{CC}_j \tag{24.26}$$

where $A_{b,j}$ (km$^2$ s$^{-1}$) is burned area for the $j$th PFT; $\mathbf{C_j} = (C_{leaf}, C_{stem}, C_{root}, C_{ts})$ is a vector with carbon density (g C km$^{-2}$) for leaf, stem (live and dead stem), root (fine, live coarse and dead coarse root), and transfer and storage carbon pools as elements; $\mathbf{CC}_j = (\mathbf{CC}_{leaf}, \mathbf{CC}_{stem}, \mathbf{CC}_{root}, \mathbf{CC}_{ts})$ is the corresponding combustion completeness factor vector (Table 24.1). Moreover, we assume that 50% and 28% of column-level litter and coarse woody debris are burned and the corresponding carbon is transferred to atmosphere.

Tissue mortality due to fire leads to carbon transfers in two ways. First, carbon from uncombusted leaf, live stem, dead stem, root, and transfer and storage pools $\mathbf{C}'_{\mathbf{j1}} = (C_{leaf}(1 - CC_{leaf}), C_{livestem}(1 - CC_{stem}), C_{deadstem}(1 - CC_{stem}), C_{root}(1 - CC_{root}), C_{ts}(1 - CC_{ts}))_j$ (g C km$^{-2}$) is transferred to litter as

$$\Psi_{j1} = \frac{A_{b,j}}{f_j A_g}\mathbf{C}'_{\mathbf{j1}} \bullet M_{j1} \tag{24.27}$$

L96 (Eq. 1), L106 (Eq. 2): All vectors should be in boldface italics: https://www.geos cientific-model-development.net/submission.html#math

>> We have corrected them in the revised manuscript.

L114: Delete "Especially, the " "represent" should be "represents"

>> We have corrected them in the revised manuscript.

L127: "leaf size" should be "leaf area"

>> We have corrected them in the revised manuscript.

L141–145 Is Veraverbeke et al. (2015) the citation for AKFED? If so, cite that in the first sentence here. If not, add the correct citation—and then why is Veraverbeke et al. (2015) discussed at all?

>> Yes, Veraverbeke et al. (2015) is the citation for AKFED. We have corrected it in the revised manuscript.

L170: *We also used data on Alaskan carbon emissions from the AKFED (Veraverbeke et al., 2015) to evaluate the model performance for carbon emissions in Alaska (Table 1).*

L144: "presumed" is almost certainly not the right word. "Calculated"? "Determined"?

>> We have changed it into "estimated".

L174: *They estimated that the highest carbon emission was 69 Tg C in 2004 and the annual carbon emission was 15 Tg C.*

L146: Not just EXP-GFED4, right? Also OL?

>> Yes, that is correct; we have corrected it in the revised manuscript.

L169: Replace "but" with "and"

>> We revised this in the revised manuscript.

L195: Replace "big fires" with ""large burned areas or "anomalous years"; "big fires" implies individual contiguous burn patches, which may not be the case.

>> We revised this in the manuscript.

L196: *Studies suggested that these large burned areas were associated with a high lightning frequency and drought*

L199: What is this sentence trying to say? Why "especially"? What's special about it?

>> We meant that the simulated burned area in Eastern Siberia increased with time (2001–2012), which is not shown in GFED. To clarify this, we have re-written the sentence in the revised manuscript.

L200: *Although the GFED burned area in Eastern Siberia did not vary significantly over time, the simulated burned area increased from 2001 to 2012 at a rate of 0.33 Mha/year.*

Paragraph at L201–5 needs a total rework.

>> As per the editor's suggestion, we have re-written the paragraph in the revised manuscript.

L203: *Figure 4 shows the spatial distribution of the burned areas of GFED4 and CLM-Default in 2004 over Alaska. The number of grid cells (0.5 × 0.5 degree) in CLM-Default where the burned areas exceeded 0.1 ha in 2004 was more than 50. In contrast to the GFED4 burned areas, there were two grid cells with more than 0.1 ha of burned areas simulated using CLM5-BGC in Alaska (Table 2). Table 2 shows that CLM5-BGC has a limitation in simulating large burned areas in Alaska. Small fires were simulated in more grid cells, and the simulated burned areas were more widely distributed than those in the GFED4 products.*

"inadequately" is a value judgment; whether the model performs adequately depends on what question it's being used to answer. Replace this with something that describes the CLM bias objectively.

>> We have removed it, as we have re-written the paragraph according to the previous suggestion.

L201: Re-state grid cell resolution here.

>> We have added the grid-cell resolution in the revised manuscript.

L204: *The number of grid cells (0.5 × 0.5 degree) in CLM-Default where the burned areas exceeded 0.1 ha in 2004 was more than 50.*

L202: Observed by GFED? Or AKFED?

>> The sentence has been deleted in the revised manuscript.

L202–3:"a few"? How many?

>> We have clarified it in the revised manuscript.

L 204: *In contrast to the GFED4 burned areas, there were two grid cells with more than 0.1 ha of burned areas simulated using CLM5-BGC in Alaska (Table 2).*

L204: "simulating largely burned areas"? What does this mean?

>> We have clarified it in the revised manuscript.

L205: *Table 2 shows that CLM5-BGC has a limitation in simulating large burned areas in Alaska.*

L204: "more grid cells"? Relative to what?

>> We have clarified it in the revised manuscript.

L206: *Small fires were simulated in more grid cells, and the simulated burned areas were more widely distributed than those in the GFED4 products.*

Paragraphs at L207–26 need rework.

Please combine these paragraphs. You do some discussion in the second paragraph, which is confusing because the paragraph break makes it seem like you're moving on to something else.

>> As per the editor's suggestion, we have re-written these paragraphs and moved them to the discussion section in the revised manuscript.

L304: *Difference in burned area between the model and observation may be attributed to incorrect input data such as lightning frequency and fire management as well as a misrepresentation of fire processes. First, the limited representation of fire ignition sources and spread may create discrepancies between modeled and observed burned areas. Lightning, which is a major source of fire at high latitudes, especially in Alaska, has increased because of the warming climate (Kępski and Kubicki, 2022). Although the lightning frequency at high latitudes varied yearly, the climatology of the 3-hourly lightning frequency from 1995 to 2011 was used in CLM. Moreover, the calculated ratio of cloud-to-ground lightning has large*

*uncertainties and may cause models to misestimate fire ignition and burned areas. Furthermore, it is inherent that the grid-based large-scale model is limited in capturing micro-environmental impacts on fire spread. Fires spread differs depending not only on the temperature, precipitation, wind speed, and direction but also on the composition of vegetation at the local scale.*

*In addition, wildfires are strongly affected by the weather conditions after the fire ignition. For example, wind and precipitation determine the spread and duration of fire. However, in CLM5-BGC, the fire ignition and fire spread rate are simultaneously calculated based on the weather conditions of fire ignition or pre-fire. Moreover, wildfires in ecosystems persist from hours to months, depending on ecosystem characteristics and climate conditions. However, the duration of each fire is assumed to be equal to one day in CLM5-BGC (Li et al., 2012). For example, Andela et al. (2019) reported that the average fire duration in a boreal forest was longer than those in other regions, and the average size of each fire in the boreal forest was larger than those in temporal forests and under deforestation. Moreover, wind speed is an important factor determining fire spread in the model. In CLM, the spread of fire increases as the wind speed increases. However, according to Lasslop et al. (2015), there is strong variation in the burned fraction with wind speed, characterized by an increase until a certain wind speed threshold is reached and a decrease thereafter. The study suggests that global fire models should avoid a strong amplification for higher wind speeds to prevent overestimation of modelled burned areas.*

*The management system and infrastructures for fires vary by country or region. For instance, there are four types of fire policy options in Alaska, namely critical, full, modified, and limited, according to the levels of anthropogenic effort in extinguishing the fire (Phillips et al., 2022). For example, fire suppression is the highest priority at the critical protection level because wildfire can threaten human life and inhabited property. The lowest priority for fire-related resource assignments is applied at the limited protection level. In Alaska, areas under the full, modified, and limited management options occupy 16%, 16%, and 67% of Alaska, respectively. Critical-protection-level areas occupy less than 1% of Alaska. In CLM5-BGC, however, the suppression impact is calculated based on the GDP and population, which may underestimate burned areas in the limited regions of Alaska because of the large GDP of the United States.*

You should also discuss the issues with wind speed in global fire models: Lasslop et al. (2015), https://www.publish.csiro.au/wf/WF15052

>> We have added this issue in the discussion section in the revised manuscript.

L314: *In CLM, the spread of fire increases as the wind speed increases. However, according to Lasslop et al. (2015), there is strong variation in the burned fraction with wind speed, characterized by an increase until a certain wind speed threshold is reached and a decrease thereafter. The study suggests that global fire models should avoid a strong amplification for higher wind speeds to prevent overestimation of modelled burned areas.*

"position"?

>> The related sentence has been re-written, and the word has been omitted from the revised manuscript.

"misunderstanding" is not the right word. Do you mean "misrepresentation"?

>> We changed the word to "the limited representation"

What do you mean by "the limitation of using point data in the grid-based model"?

>> It has been clarified in the revised manuscript.

L310: *Furthermore, it is inherent that the grid-based large-scale model is limited in capturing micro-environmental impacts on fire spread. Fires spread differs depending not only on the temperature, precipitation, wind speed, and direction but also on the composition of vegetation at the local scale.*

L219: "Persistence" ("duration" would be clearer) has what units?

>>We have added the unit in the manuscript.

L310: *However, the duration of each fire is assumed to be equal to one day in CLM5-BGC (Li et al., 2012).*

L220: "fires can last longer" in CLM or real life?

>> The sentence has been omitted from the revised manuscript.

Expand discussion of fire duration into its own paragraph and add citations. Much literature exists about both (a) real-world fire durations (especially in Alaska, where large fires contribute a huge proportion of burned area), (b) the effect of the constant-duration (or max 1 day) assumption in fire models, and (c) the effect of including dynamic, > 1 day fire duration in models.

>> As per the editor's suggestion, we have added an explanation in the revised manuscript

L310: *In addition, wildfires are strongly affected by the weather conditions after the fire ignition. For example, wind and precipitation determine the spread and duration of fire. However, in CLM5-BGC, the fire ignition and fire spread rate are simultaneously calculated based on the weather conditions of fire ignition or pre-fire. Moreover, wildfires in ecosystems persist from hours to months, depending on ecosystem characteristics and climate conditions. However, the duration of each fire is assumed to be equal to one day in CLM5-BGC (Li et al., 2012). For example, Andela et al. (2019) reported that the average fire duration in a boreal forest was longer than those in other regions, and the average size of each fire in the boreal forest was larger than those in temporal forests and under deforestation. Moreover, wind speed is an important factor determining fire spread in the model. In CLM, the spread of fire increases as the wind speed increases. However, according to Lasslop et al. (2015), there is strong variation in the burned fraction with wind speed, characterized by an increase until a certain wind speed threshold is reached and a decrease thereafter. The study suggests that global fire models should avoid a strong amplification for higher wind speeds to prevent overestimation of modelled burned areas.*

Paragraph about Alaskan fire policy needs expansion. What do those different levels mean? How much area is in each level, especially in your study area?

>> In Alaska, areas are divided by priority for firefighting. For example, firefighting in critical areas takes precedence over those in other areas. In Alaska, areas under the full, modified, and limited management options occupy 16%, 16%, and 67% of Alaska, respectively. We have added these in the revised manuscript.

L324: *The management system and infrastructures for fires vary by country or region. For instance, there are four types of fire policy options in Alaska, namely critical, full, modified, and limited, according to the levels of anthropogenic effort in extinguishing the fire (Phillips*

*et al., 2022). For example, fire suppression is the highest priority at the critical protection level because wildfire can threaten human life and inhabited property. The lowest priority for fire-related resource assignments is applied at the limited protection level. In Alaska, areas under the full, modified, and limited management options occupy 16%, 16%, and 67% of Alaska, respectively. Critical-protection-level areas occupy less than 1% of Alaska. In CLM5-BGC, however, the suppression impact is calculated based on the GDP and population, which may underestimate burned areas in the limited regions of Alaska because of the large GDP of the United States.*

L224: "anthropophonic" is not a word. "anthropogenic"?

>> We have corrected it.

It's unclear what the difference is between Sections 4.2 and 4.3. You should strongly consider combining them to tell a more cohesive story about your results.

>> In section 4.2, we explained the difference in carbon fluxes between the two simulations. Therefore, we have combined the two sections and moved a few paragraphs to the discussion section.

L241–9

Did C emissions change much between what Veraverbeke et al. looked at (GFED3s) and GFED4?

>> We have deleted a few sentences about GFED3 because we consider them unnecessary.

Be clearer throughout about when you're discussing CLM vs. GFED (vs. real life?) combustion completeness factors.

>> We have deleted a few sentences about GFED3 because we consider them unnecessary. In addition, we have clarified our meaning regarding the combustion completeness factors in the revised manuscript, as shown below.

L430: *The combustion completeness factor for leaves is 0.8 and that for stems ranges from 0.27–0.8, depending on the PFTs in CLM5-BGC. According to van der Werf et al. (2010), the*

*combustion completeness factor of aboveground live biomass, which ranges from 0.3–0.4 in the boreal region, is lower than that in other regions. Therefore, the combustion completeness factors for boreal trees may be lower than the current default value in CLM5-BGC.*

*The carbon emission simulation was highly improved after replacing the fire simulation with GFED4 in Eastern Siberia (Figure 5b); the correlation was improved from 0.41 in CLM-Default to 0.88 in EXP-GFED4, and the RMSE was reduced from 19.74 g m$^{-2}$ year$^{-1}$ in CLM-Default to 4.2 g m$^{-2}$ year$^{-1}$ in EXP-GFED4, compared with the GFED4 products. In Eastern Siberia, grasses are dominant, suggesting that the value of the combustion completeness factors for grass in CLM5-BGC is more similar to those of GFED4 products than to those of boreal trees.*

You cite the combustion completeness factors for GFED3 (van der Werf et al., 2010) instead of the dataset you actually used (GFED4; Giglio et al., 2013). There were actually important changes to how combustion completeness works in GFED4!

>> GFED (v3 and v4) emissions are derived from the Carnegie-Ames-Stanford Approach (CASA) biosphere model. In the model, the metrics for combustion completeness (CC) are used to calculate emissions (Seiler and Crutzen, 1980). Therefore, we have cited van der Werf et al. (2010) in the manuscript. van der Werf et al., (2010) reported on the value of combustion completeness factors, but there is little mention of the value of combustion completeness factors in Giglio et al. (2013).

L248: Tilde should be an en dash

>> We have corrected it in the revised manuscript.

L251: "form" should be "from"

>> We have corrected it in the revised manuscript.

L253:

"more reliable" in what? GFED/CASA or CLM?

>> The value of the combustion completeness factors for grass in CLM5-BGC is more similar

to those of GFED4 products. We have clarified this in the revised manuscript.

L437: *In Eastern Siberia, grasses are dominant, suggesting that the value of the combustion completeness factors for grass in CLM5-BGC is more similar to those of GFED4 products than to those of boreal trees.*

"dominant" in what? Observations and/or GFED/CASA and/or CLM?

>> We have deleted this word because we found it unnecessary.

L256–61: This paragraph feels weird in a section about carbon fluxes without you first having discussed GPP/NEP/NPP. LAI is an explanatory factor of those things and thus should go after the GPP/NEE/NEP discussion.

>> As per the editor's suggestion, we have moved the explanation on LAI after that on GPP/NPP/NEP.

L238: *Unlike carbon emissions, the regionally-averaged GPP, NPP, and NEP (Fig. 6c–6h) did not significantly change in EXP-GFED4. The differences in GPP, NPP, and NEP are less than 3%, indicating that fires rarely impacted carbon fluxes related to vegetation and decomposition. This is because the ratio of the fire area to the total area was relatively small. For example, the highest annual burned area of all simulations was 6 Mha, which accounted for 6.87% of our study domain. The simulated LAIs in Alaska and Eastern Siberia are presented in Fig. 6a and 6b, respectively. In Alaska (Fig. 6a), the difference in LAI between CLM-Default and EXP-GFED4 was the largest in 2005 (0.03 m2/m2). Although the difference in burned area between CLM-Default and GFED4 (Fig. 3a) was the largest in 2004, the largest difference in LAI was in 2005 since vegetation damage caused by fire in 2004 had not fully recovered, and the difference in burned area in 2005 was also quite large. In Eastern Siberia (Fig. 6b), the difference in the simulated LAI between CLM-Default and EXP-GFED4 has been large since 2009, when the difference in the size of burned areas was amplified (Fig. 3b). Although the LAI, which affects primary GPP and other carbon fluxes, was reduced by fires, the LAI after fires was not substantially different owing to the small fire area compared to the total area.*

L264: "rate" is not correct here, as it implies something with time in the denominator. Replace "rates of changes" with "differences".

>> We have corrected them.

L290–6: This paragraph fits more in the Conclusions section.

>> We have moved one sentence of this paragraph to the conclusions section and deleted others because of overlap.

L321–31:

At some point this paragraph transitions from talking about both regions to just Alaska. Make Alaska its own paragraph, as you did for Siberia.

>> We have divided the paragraph into two paragraphs: one explaining both regions and the other explaining Alaska, with additional sentences.

L321: *To investigate the fire impacts on water fluxes, we compared the results of ET and ET components, such as canopy evaporation, canopy transpiration, and ground evaporation, in six grid cells where the differences in burned area between CLM-Default and EXP-GFED4 are the largest in Alaska and Eastern Siberia (Figure 8). Because the LAI decreases owing to wildfires, canopy evaporation and canopy transpiration decrease in the burned areas.*

*We observed that more rainfall reaches the ground, which would make the ground evaporation rate higher in regions with more burned areas, especially in 2004 and 2005 in Alaska. The differences in annual canopy evaporation, canopy transpiration, and ground evaporation between the two simulations were 5.41 mm and 13.37 mm, 2.3 mm and 6.26 mm, and −1.39 mm and −7.4 mm in 2004 and 2005, respectively. Canopy transpiration decreased by 3%, canopy evaporation decreased by 12%, and ground evaporation increased by 10% in 2004 and 2005 after applying the GFED4 burned area into CLM. This is consistent with the findings of Li et al. (2017) and Seo and Kim (2019), showing that canopy evaporation and canopy transpiration decreased, and ground evaporation increased when comparing the simulation with and without fire. Furthermore, the total ET in the presence of fire decreased by 6.32 mm and 12.08 mm in 2004 and 2005, respectively, indicating that canopy evaporation is more strongly influenced by fires over Alaska in CLM.*

L322: Replace "grids" with "grid cells"

>> We have corrected this in the revised manuscript.

L323: "affected" in what direction?

>> We have clarified it in the revised manuscript.

L323: *Because the LAI decreases owing to wildfires, canopy evaporation and canopy transpiration decrease in the burned areas.*

L324: "may"?

>> We have deleted the word in the revised manuscript.

L329: Replace "the CLM-dynamic global vegetation model" with just "CLM"

>> We have corrected it in the revised manuscript.

L370: Please cite the specific version (git tag or commit SHA) of CLM on which you made your changes.

>> We have added the git tag in the revised manuscript.

L465–6: Please replace citation with Rabin et al. (2018): https://gmd.copernicus.org/articles/11/815/201 8/

>> We have replaced the citation in the revised manuscript.

Figs. 2, 3, 5, 6, 9, 10: Please use more colorblind-friendly colors in Fig. 5, especially avoiding red and green. For all these figures, using different line styles (solid vs. dashed vs. dotted) and/or a variety of markers (instead of just squares) would also help. Some useful resources can be found here: https://www.geoscientific-model-development.net/submission.html#figurestables

>> We have changed the figure colors and styles in the revised manuscript, as shown below.

[Figure]

**Figure 1. Flow diagram for CLM-Default (orange line) and EXP-GFED4 (blue line).**

CLM-Default, default CLM5-BGC simulation; EXP-GFED4, experimental simulation with global fire emission database

[Figure]

**Figure 2. Burned area based on GFED4 and simulated burned area of CLM-Default over (a) Alaska (b) and Eastern Siberia from 2001 to 2012.**

GFED4, global fire emission database (version 4); OL, CLM-Default, default CLM5-BGC simulation

[Figure]

**Figure 3. Simulated carbon fluxes of CLM-Default and EXP-GFED4 such as carbon emission (a,b), and (c,d) in Alaska (a,c) and Eastern Siberia (b,d) from 2001 to 2012. GFED carbon emission (a,b) and AKFED carbon emission (a) are added to evaluate the performance of carbon emission in CLM-Default and EXP-GFED4 runs. Also, NEE of GEOS-Carb CASA-GFED was added to evaluate the performance of NEE in CLM-Default and EXP-GFED4 runs (c, d).-**

CLM-Default, default CLM5-BGC simulation; EXP- GFED4, experimental simulation with global fire emission database (version 4); AKFED; Alaskan Fire Emissions Database; NEE, net ecosystem exchange

[Figure]

**Figure 4. Simulated LAI (a, b), and carbon fluxes of CLM-Default and EXP-GFED4 such as GPP (c, d), NPP (e, f), and NEP (g, h) in Alaska (a, c, e, g) and Eastern Siberia (b, d, f, h) from 2001 to 2012.**

LAI, leaf area index; CLM-Default, default CLM5-BGC simulation; EXP-GFED4, experimental simulation with global fire emission database (version 4); GPP, gross primary production; NPP, net primary production; NEP, net ecosystem production

[Figure]

**Figure 5. Simulated burned area (a, b), and water fluxes of CLM-Default and EXP-GFED4 such as ET (c, d), ground evaporation (GE; e, f), canopy evaporation (CE; g, h), and canopy transpiration (CT; i, j) in five grids where the difference in burned area between CLM-Default and EXP-GFED4 is highest in Alaska (a, c, e, g, i) and Eastern Siberia (b, d, f, h, j) from 2001 to 2012.**

CLM-Default, default CLM5-BGC simulation; EXP-GFED4, experimental simulation with global fire emission database (version 4); ET, evapotranspiration

[Figure]

**Figure 6. Differences (the value of OL- the value of EXP-GFED4) in simulated top soil (0–20 cm) moisture and bottom soil (70–150 cm) moisture in Alaska (a) and Eastern Siberia (b).**

CLM-Default, default CLM5-BGC simulation; EXP-GFED4, experimental simulation with global fire emission database (version 4)

---

## Author Response (AR3)

**Response Letter**

**Editor:**

Public justification (visible to the public if the article is accepted and published):

Thank you for your revisions so far. I have a few more requests that I can review myself before acceptance:

- My question about land use areas was specifically about land *use*. Did your spinup include cropland or no?

>> Yes, the spinup included the cropland. We have added this in the revised manuscript.

L148: *Because starting a new simulation at a different spatial resolution could introduce model artifacts, we ran CLM5-BGC at a 0.5°×0.5° spatial resolution from the initial state, including the land use, such as cropland, for 200 years for the equilibration with repeatedly using Climate Research Unit (CRU) – National Centers for Environmental Prediction (NCEP) reanalysis climate data for 1980-2000.*

- Please replace citations of equations in CLM Technical Description with their equivalent citations from Li et al. (2012, Biogeosciences: https://bg.copernicus.org/articles/9/2761/2012/)

>> As per editor's suggestion, we have replaced the citation in the revised manuscript.

L93: *The PFT-level carbon emission from the fire is calculated as follows (Li et al., 2012):*

L102: *In CLM5-BGC, the amount of leaf carbon to litter ($\Psi$) caused by fire is calculated as follows (Li et al., 2012):*

L149-154, "spinup" section:

- It's confusing to refer to this as a "spinup", as you saw with one of the reviewers. Maybe "equilibration" would be better? It would help to mention that the year-2000 initial conditions

are the result of a complete spinup.

- "was repeatedly run for 200 years" implies some number of 200-year runs. I think "repeatedly" refers to the repetition of the 1980-2000 climate forcing (which is actually 21 years, not 20).

>> As per editor's suggestion, we have used "equilibration" and clarified the data period in the revised manuscript.

L148: *Because starting a new simulation at a different spatial resolution could introduce model artifacts, we ran CLM5-BGC at a 0.5°×0.5° spatial resolution from the initial state, including the land use, such as cropland, for 200 years for the equilibration with repeatedly using Climate Research Unit (CRU) – National Centers for Environmental Prediction (NCEP) reanalysis climate data for 1980-2000.*

- Despite the text in this paragraph, Figure 2 doesn't appear to have anything to do with the spinup. In fact, Figure 2 is generally confusing. Why are arrows leading from atmospheric forcing and surface condition to GFED4 burned area, implying those are somehow inputs in your workflow to GFED4?

>> We have revised the Figure 2 to clearly describe the different experiments of this study.

[Figure]

**Figure 1. Flow diagram for CLM-Default and EXP-GFED4.**

- This paragraph should mention the time period of your experimental runs.

- Suggested rewrite of this part of the text: "Our simulations started with a pre-existing initial

condition state for 2000 at XXX resolution. Because starting a new run at our 0.5° resolution could introduce model artifacts, we ran CLM-Default from this state for 200 years using repeated YYYY–ZZZZ CRU-NCEP forcing data before starting our experiments with transient 2000-2012 climate."

>> As editor's suggestion, we have revised the paragraph for the experimental design with correcting the data period.

L144: *Figure 2 shows the experimental process of this study. Our simulations started with a pre-existing initial condition state for the year 2000 at a 1.9° × 2.5° spatial resolution provided by NCAR. Because starting a new simulation at a different spatial resolution could introduce model artifacts, we ran CLM5-BGC at a 0.5° × 0.5° spatial resolution from the initial state, including the land use, such as cropland, for 200 years for the equilibration with repeatedly using Climate Research Unit (CRU) – National Centers for Environmental Prediction (NCEP) reanalysis climate data for 1980-2000. Then, CLM-Default and EXP-GFED4 were simulated for 12 years (2001-2012) at the 0.5° × 0.5° spatial resolution using CRU-NCEP atmospheric forcing, which include precipitation, temperature, wind speed, surface pressure, specific humidity, longwave radiation, and solar radiation.*

L205-9 and the related Fig. 4 still need some rework:

- "The number of grid cells... in CLM-Default where the burned areas exceeded 0.1 ha in 2004 was more than 50." This should be GFED4 instead of CLM Default.

>> We have corrected the number from 0.1 to 0.01 Mha and replaced "CLM Default" with "GFED4" in the revised manuscript.

- "In contrast to the GFED4 burned areas" should just be "In contrast"

>> We have corrected it in the revised manuscript.

Fig. 4:

- It seems like some gridcells exceed the highest value in the colorbars. This should be indicated on the colorbars with a dark triangle at the top and/or a ≥ symbol in front of 0.01.

- Yellow on white is very hard to see. Please use a different color scale, or consider a gray background to this figure.

>>As editor's suggestion, we have added the ≥ symbol and changed the background color to gray in Figure 4.

[Figure]

**Figure 2. Spatial distribution of burned area of (a) GFED4 (b) and CLM-Default in 2004 over Alaska.**

GFED4, global fire emission database (version 4); CLM-Default, default CLM5-BGC simulation

Minor comments:

- Throughout: Replace "leaf size" with "leaf area"

>> We have corrected them in the revised manuscript.

- Figures:

- "OL" still in Fig. 4 and captions of Figs. 3 and 9. Replace with CLM Default.

>> We have corrected them in the revised manuscript.

- Please consider adding

- L13: Rename "open-loop" to match rest of manuscript

>> We have corrected them in the revised manuscript.

L12: *The results showed that the simulated carbon emissions with burned areas from GFED4 (i.e., experimental run) were significantly improved in comparison to the default CLM5-BGC simulation, which resulted in opposite signs of the net ecosystem exchange for 2004, 2005, and 2009 over Alaska between the default and experimental runs.*

- L281: Negative sign needed in parenthetical at "–9.19 mm (28%)".

>> We have added the negative sign in the revised manuscript.

- L293: Delete "$f$"

>> We have deleted it.